# MULTILINEAR OPERATOR NETWORKS

**Yixin Cheng[1], Grigorios G. Chrysos[2], Markos Georgopoulos, Volkan Cevher[1]**
[1]LIONS - École Polytechnique Fédérale de Lausanne     [2]University of Wisconsin-Madison

## ABSTRACT

Despite the remarkable capabilities of deep neural networks in image recognition, the dependence on activation functions remains a largely unexplored area and has yet to be eliminated. On the other hand, Polynomial Networks is a class of models that does not require activation functions, but have yet to perform on par with modern architectures. In this work, we aim close this gap and propose MONet, which relies *solely* on multilinear operators. The core layer of MONet, called Mu-Layer, captures multiplicative interactions of the elements of the input token. MONet captures high-degree interactions of the input elements and we demonstrate the efficacy of our approach on a series of image recognition and scientific computing benchmarks. The proposed model outperforms prior polynomial networks and performs on par with modern architectures. We believe that MONet can inspire further research on models that use entirely multilinear operations. The source code is available at MONet.

## 1 INTRODUCTION

Image recognition has long served as a crucial benchmark for evaluating architecture designs, including the seminal ResNet (He et al., 2016) and MLP-Mixer (Tolstikhin et al., 2021). As architectures are applied to new applications, there are additional requirements for the architecture design. For instance, encryption is a key requirements in safety-critical applications (Caruana et al., 2015). Concretely, the Leveled Fully Homomorphic Encryption (LFHE) (Brakerski et al., 2014), can provide a high level of security for sensitive information. The core limitation of FHE (and especially LFHE) is that they support only addition and multiplication as operations. That means that traditional neural networks cannot fit into such privacy constraints owing to their dependence on elementwise activation functions, making developments in MLP-Mixer and similar models invalid for many real-world applications. Therefore, new designs that can satisfy those constraints and still achieve high accuracy on image recognition are required.

A core advantage of Polynomial Nets (PNs), that express the output as high-degree interactions between the input elements (Ivakhnenko, 1971; Shin & Ghosh, 1991; Chrysos et al., 2020), is that they can satisfy constraints, such as encryption or interpretability (Dubey et al., 2022). However, a major drawback of PNs so far is that they fall short of the performance of modern architectures on standard machine learning benchmarks, such as image recognition. This is precisely the gap we aim to close in this work.

We introduce a class of PNs, dubbed Multilinear Operator Network (MONet), which is based solely on multilinear operations[1]. The core layer captures multiplicative interactions within the token elements[2]. The multiplicative interaction is captured using two parallel branches, each of which assumes a different rank to enable different information to flow. By composing sequentially such layers, the model captures high-degree interactions between the input elements and can predict the target signal, e.g., class in the case of image recognition.

Concretely, our contributions can be summarized as:

---

[1]The terminology on multilinear operations arises from the multilinear algebra. Concretely, we follow the terminology of the seminal paper of Kolda & Bader (2009).

[2]Consistent with the recent literature of MLP-based models and transformers (Dosovitskiy et al., 2021), we consider sequences of tokens as inputs. In the case of images, a token refers to a (vectorized) patch of the input image.

- We propose Mu-Layer, a new module which uses purely multilinear operations to capture multiplicative interactions. We showcase how this module can serve as a plug-in replacement to standard MLP.

- We introduce MONet, which captures high-degree interactions between the input elements. To our knowledge, this is the first network that obtains a strong performance on challenging benchmarks.

- We conduct a thorough evaluation of the proposed model across standard image recognition benchmarks to show the efficiency and effectiveness of our method. MONet significantly improves the performance of the prior art on polynomial networks, while it is on par with other recent strong-performing architectures.

The source code is available at MONet. We hope the code can enable further improvement of models relying on linear projections.

## 2 RELATED WORK

We present a brief overview of the most closely related categories of MLP-based and polynomial-based architectures from the vast literature of deep learning architectures. For a detailed overview, the interested reader can find further information on dedicated surveys on image recognition (Lu & Weng, 2007; Plested & Gedeon, 2022; Peng et al., 2022).

**MLP models**: The resurgence of MLP models in image recognition is an attempt to reduce the computational overhead of transformers (Vaswani et al., 2017). MLP-based models rely on linear layers, instead of convolutional layers or the self-attention block. The MLP-Mixer (Tolstikhin et al., 2021) is among the first networks that demonstrate a high accuracy on challenging benchmarks. The MLP-Mixer uses tokens (i.e., vectorized patches of the image) and captures both inter- and intra-token correlations. Follow-up works improve upon the simple idea of token-mixing MLP structure (Touvron et al., 2022; Liu et al., 2021). Concretely, ViP (Hou et al., 2022), cycleMLP (Chen et al., 2022), S2-MLPv2 (Yu et al., 2022a) design strategies to improve feature aggregation across spatial positions. Our work differs from MLP-based models, as it is inspired by the idea of capturing high-order interactions using polynomial expansions.

**Polynomial models**: Polynomial expansions establish connections between input variables and learnable coefficients through addition and multiplication. Polynomial Nets (PNs) express the output variable (e.g., class) as a high-degree expansion of the input elements (e.g., pixels of an image) using learnable parameters. Even though extracting such polynomial features is not a new idea (Shin & Ghosh, 1991; Li, 2003; Dauphin et al., 2017), it has been largely sidelined from architecture design.

The last few years PNs have demonstrated promising performance in standard benchmarks in various vision tasks including image recognition (Chrysos et al., 2022). In particular, PNs augmented with activation functions, which are referred to as *hybrid models* in this work, can achieve state-of-the-art performance (Hu et al., 2018; Li et al., 2019; Yin et al., 2020; Babiloni et al., 2021; Yang et al., 2022; Georgopoulos et al., 2020; Chrysos et al., 2021; Georgopoulos et al., 2021). The work of Chrysos et al. (2022) introduces a taxonomy for a single block of PNs based on the degree of interactions captured. This taxonomy allows the comparison of various approaches based on a specific degree of interactions. Building upon this work, researchers have explored ways to modify the degree or type of interactions captured to improve the performance. For example, Babiloni et al. (2021) reduce computational cost of the popular non-local block (Wang et al., 2018) by capturing exactly the same third degree interactions, while Chrysos et al. (2022) investigate modifications to the degree of expansion.

Arguably, the works most closely related to ours are Chrysos et al. (2020; 2023). In Π-Nets (Chrysos et al., 2020), three different models are instantiated based on separate manifestations of three tensor decompositions. That is, the paper establishes a link between a concrete tensor decomposition and its assumptions to a concrete architecture that can be implemented in standard frameworks. The authors evaluate those different architectures and notice that they perform well, but do not surpass standard networks, such as ResNet. Chrysos et al. (2023) improve upon the Π-Nets by introducing carefully designed (implicit and explicit) regularization techniques, which reduce the overfitting of high-degree polynomial expansions.

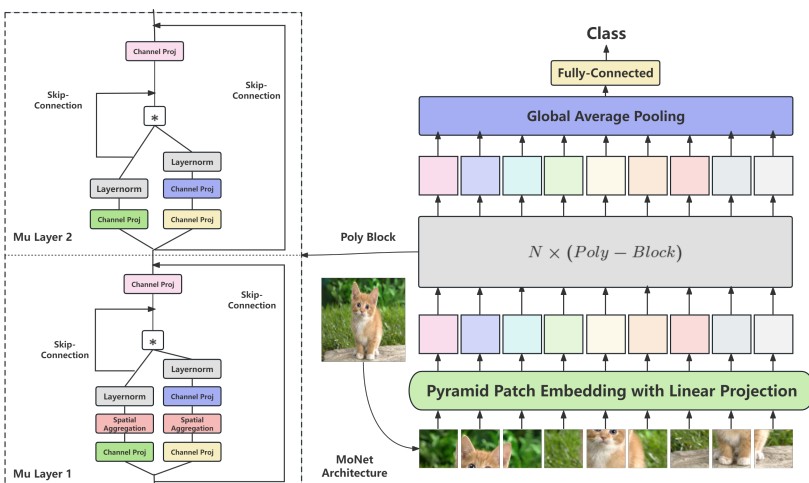

Figure 1: The architecture of the proposed Mu-Layer (on the left) and MONet (on the right). In the left figure, the grey box represents layer normalization. The color solid line boxes represent channel projection in different dimensions, all projection operations are linear. The $*$ box denotes an elementwise (Hadamard) product. The red dash box represents the spatial aggregation module.

Contrary to the aforementioned PNs, we are motivated to introduce a new architecture using PN that is comparable to modern networks. To achieve that, we are inspired by the modern setup of considering the input as a sequence of tokens. The token-based input is widely used across a range of domains and modalities that last few years. The token-based input also departs from the design of previous PNs that utilize convolutional layers instead of simple matrix multiplications. Arguably, our choice results in a weaker inductive bias as pointed out in the related works of MLP-based models. This can be particularly useful in domains outside of images, e.g., in the ODE experiments.

## 3 METHOD

Let us now introduce MONet, which expresses the output as a polynomial expansion of the input using multilinear operations. Firstly, we introduce the core layer, called Mu-Layer, in Section 3.1. Then, in Section 3.2, we design the whole architecture with a schematic visualized in Fig. 1.

**Tokens**: Following the conventions of recent MLP-based models, we consider an image as a sequence of tokens. In practice, each token is a patch of the image. We denote the sequence of tokens as $\boldsymbol{X} \in \mathbb{R}^{d \times s}$, where $d \in \mathbb{N}$ is the length of a token and $s \in \mathbb{N}$ is the number of tokens. As such, our method below assumes an input sequence $\boldsymbol{X}$ is provided. MLP-based models capture linear interactions between the elements of a token. That operation would be denoted as $\boldsymbol{\Gamma X}$, where $\boldsymbol{\Gamma} \in \mathbb{R}^{o \times d}$ is a learnable matrix.

### 3.1 MU-LAYER

Our goal is to capture rich interactions among the input elements of a token. We can achieve that using multiplicative interactions. Specifically, we use two branches, and each captures a set of linear interactions inside the token. An elementwise product is then used to capture the multiplicative interactions between the branches. Notation-wise, the output is expressed as $(\boldsymbol{AX}) * (\boldsymbol{\Lambda X})$, where $*$ denotes the elementwise product and $\boldsymbol{A}, \boldsymbol{\Lambda}$ are learnable matrices.

We perform the following three modifications to the aforementioned block. Firstly, we add a shortcut connection to capture the first-degree interactions. Secondly, we decompose $\boldsymbol{\Lambda}$ into two matrices as $\boldsymbol{\Lambda} = \boldsymbol{BD}$. This rank factorization of $\boldsymbol{\Lambda}$ enables us to capture low-rank representations in this branch. Thirdly, we add one matrix $\boldsymbol{C}$ to capture the linear interactions of the whole expression. Then, the complete layer is expressed as follows:

$$\boldsymbol{Y} = \boldsymbol{C}\left[(\boldsymbol{AX}) * (\boldsymbol{BDX}) + \boldsymbol{AX}\right] \ , \tag{1}$$

Table 1: Configurations of different MONet models. We present four models: two variants with different parameter-size (Tiny, Small) and two variants with different hidden-size. The original model use the same hidden-size across the architecture. The Multi-stage model adopts a different hidden sizes across the architecture. The number in stages represents the number of blocks that each stage includes, while the hidden size list numbers correspond to each stage's hidden size. We provide further details along with a schematic in Appendix B. We only use Multi-stage models for high-resolution image classification benchmarks, e.g., on ImageNet1K.

| Specification | Tiny | Small | Multi-stage-Tiny | Multi-stage-Small |
|---|---|---|---|---|
| Numbers of Blocks | 32 | 45 | 32 | 32 |
| Embedding Method | Pyramid Patch Embedding | Pyramid Patch Embedding | Pyramid Patch Embedding | Pyramid Patch Embedding |
| Hidden size | 192 | 320 | 64,128,192,192 | 128,192,256,384 |
| Stages | - | - | 4,8,12,10 | 4,6,12,14 |
| Expansion Ratio | 3 | 3 | 3 | 3 |
| Shrinkage Ratio | 4 | 4 | 8 | 8 |
| Parameters | 14.0M | 56.8M | 10.3M | 32.9M |
| FLOPs | 3.6 | 11.2 | 2.8 | 6.8 |

where $A \in \mathbb{R}^{m \times d}, B \in \mathbb{R}^{m \times l}, C \in \mathbb{R}^{o \times m}, D \in \mathbb{R}^{l \times d}$ are learnable parameters and $*$ symbolizes an elementwise product. The block has two hyperparameters we can choose, i.e., the rank $m \in \mathbb{N}$ and the value $l \in \mathbb{N}$. In practice, we utilize a shrinkage ratio $\frac{m}{l} > 1$ to encourage different information flow in the branches. Prop. 1, the proof of which is in Appendix A, indicates the interactions captured. The schematic of this layer, called Mu-Layer, is depicted in Appendix C.

**Proposition 1.** *The Mu-Layer captures multiplicative interactions between elements of each token.*

**Poly-Block**: The Poly-Block, which is the core operation of MONet architecture, stacks Mu-Layers sequentially. Concretely, we connect two Mu-Layers sequentially and add a a layer normalization (Ba et al., 2016) between them[3]. Additionally, we add a shortcut connection to skip the first block. The two Mu-Layers are similar except for two differences: (a) a spatial shift operation (Yu et al., 2022b) is added to the first block, (b) the hidden dimension is $m$ in the first Mu-Layer, and $m \cdot \epsilon$ in the second, where $\epsilon$ is an expansion factor. The Poly-Block is illustrated in Fig. 1.

## 3.2 NETWORK ARCHITECTURE

Our final architecture is composed of $N$ Poly-Blocks. Each block captures up to $4^{\text{th}}$ degree interactions, which results in the final architecture capturing up to $4^N$ interactions, with $N > 10$ in practice[4]. Beyond the Poly-Block, one critical detail in our final architecture is the patch embedding.

**Patch embeddings**: In deep learning architectures, the choice of patch size plays a crucial role in capturing fine-grained features. Smaller patches prove more effective in this regard. However, a trade-off arises, as reducing the patch size leads to an increase in floating-point operations, which scales approximately quadratically. To overcome this challenge, we introduce a novel scheme called pyramid patch embedding. This scheme operates by considering embeddings at smaller scales and subsequently extracting new patch embeddings on top. By adopting this approach, we can leverage the finer details of features without introducing additional parameters or computational complexity. Our experiments validate the effectiveness of incorporating multi-level patches, enhancing the overall performance. For the ImageNet1K dataset, we employ a two-level pyramid scheme, although it is worth noting that higher-resolution images may benefit from additional levels. We present the schematic and further details of this approach in Appendix D.

## 4 EXPERIMENTS

In this section, we conduct a thorough experimental validation of MONet. We conduct experiments on large-scale image classification in Section 4.1, fine-grained and small-scale image classification

---

[3]We also conduct experiments with batch normalization and obtain similar performance; results and analysis on Section 4.1.

[4]A theoretical proof on the degree of expansion required to tackle a specific task is an interesting problem that is left as future work.

in Section 4.2. In addition, we exhibit a unique advantage of models without activation functions to learn dynamic systems in scientific computing in Section 4.3. Lastly, we validate the robustness of our model to diverse perturbations in Section 4.4. We summarize four configurations of the proposed MONet in Table 1 with different versions of MONet. We present a schematic for our configuration in Appendix B. Due to the limited space, we conduct additional experiments in the Appendix. Concretely, we experiment on semantic segmentation in Appendix H, while we add additional ablations and experimental details on Appendices K to M.

## 4.1 ImageNet1K Classification

ImageNet1K, which is the standard benchmark for image classification, contains 1.2M images with 1,000 categories annotated. We consider a host of baseline models to compare the performance of MONet. Concretely, we include strong-performing polynomial models[5] (Chrysos et al., 2020; 2023), MLP-like models (Tolstikhin et al., 2021; Touvron et al., 2022; Yu et al., 2022b), models based on vanilla Transformer (Vaswani et al., 2017; Touvron et al., 2021) and several classic convolutional networks (Bello et al., 2019; Chen et al., 2018b; Simonyan & Zisserman, 2015; He et al., 2016).

**Training Setup**: We train our model using AdamW optimizer (Loshchilov & Hutter, 2019). We use a batch size of 448 per GPU to fully leverage the memory capacity of the GPU. We use a linear warmup and cosine decay schedule learning rate, while the initial learning rate is 1e-4, linear increase to 1e-3 in 10 epochs and then gradually drops to 1e-5 in 300 epochs. We use label smoothing (Szegedy et al., 2016), standard data augmentation strategies, such as Cut-Mix (Yun et al., 2019), Mix-up (Zhang et al., 2018) and auto-augment (Cubuk et al., 2019), which are used in similar methods (Tolstikhin et al., 2021; Trockman & Kolter, 2023; Touvron et al., 2022). Our data augmentation recipe follows the one used in MLP-Mixer (Tolstikhin et al., 2021). We do not use any external data for training. We train our model using native PyTorch training on 4 NVIDIA A100 GPUs.

We exhibit the performance of the compared networks in Table 2. Notice that our smaller model achieves a **10%** improvement over previous PNs. Our larger model further strengthens this performance gain and manages for the first time to close the gap between PNs and be on par with other recent models.

## 4.2 Additional benchmarks in image recognition

Beyond ImageNet1K, we experiment with a number of additional benchmarks to further assess MONet. We use the standard datasets of CIFAR10 (Krizhevsky et al., 2009), SVHN (Netzer et al., 2011) and Tiny ImageNet1K (Le & Yang, 2015) for image recognition. A fine-grained classification experiment on Oxford Flower102 (Nilsback & Zisserman, 2008) is conducted. Beyond the different distributions, those datasets offer insights into the performance of the proposed model in datasets with limited data[6]. The results in Table 3 compare the proposed method with other strong-performing methods. MONet outperforms the compared methods with the second strongest being the $\mathcal{R}$-PolyNets. MLP-models usually do not perform well in small datasets as indicated in Tolstikhin et al. (2021) due to their weaker inductive bias. Notably, MONet still performs better than other CNN-based polynomial models.

## 4.3 Poly Neural ODE Solver

An additional advantage of our method is the ability to model functions that have a polynomial form (e.g, in ordinary differential equations) in an interpretable manner (Fronk & Petzold, 2023). A neural ODE solver is a computational technique that models dynamic systems relying on ODEs using neural networks. Chen et al. (2018a); Dupont et al. (2019) illustrate the potential of neural ODEs to capture the behavior of complex systems that evolve over time. However, an evident limitation is that those models remain black-box which means the learned equation remains unknown after training. On the

---

[5]We note that we report results of $\Pi-$nets without activation functions from Chrysos et al. (2020) for a fair comparison. The models with activation functions are reported as 'hubrid'.

[6]Previous MLP-based models have demonstrated weaker performance than CNN-based models in such datasets. The original $S^2$MLP only reports two large models on these datasets. We use an open source $S^2$MLP code design two models with 12-14 M parameters for a fair comparison. Those models, noted as $S^2$MLP-Wide-S (hidden size 512, depth 8) and $S^2$MLP-Deep-S (hidden size 384, depth 12) are used for the comparisons.

Table 2: ImageNet1K classification accuracy (%) on the validation set for different models. In the Activation Function column, G denotes Gelu, R denotes Relu, T denotes Tanh, and P denotes RPReLU. The models with † are special polynomial models with activation functions. We mark model with up to 15M params in red color, models with 15-40M parameters with green color and model with over 40M parameters with blue . Beyond our models with Layer Normalization, we conduct an experiment with Batch Normalization and notice that the performance is similar (i.e., the model Multi-stage MONet-T-BatchNorm below).

| | | Accuracy | | | | | |
| | Extra Data | Top-1(%) | Top-5(%) | FLOPs (B) | Params (M) | Activation | Attention |
|---|---|---|---|---|---|---|---|
| **CNN-based** | | | | | | | |
| ResNet-18 (He et al., 2016) | ✗ | 69.7 | 89.0 | 1.8 | 11.0 | R | ✗ |
| ResNet-50 (He et al., 2016) | ✗ | 77.2 | 92.9 | 4.1 | 25.0 | R | ✗ |
| $A^2$Net (Chen et al., 2018b) | ✗ | 77.0 | 93.5 | 31.3 | 33.4 | R | ✓ |
| AA-ResNet-152 (Bello et al., 2019) | ✗ | 79.1 | 94.6 | 23.8 | 61.6 | R | ✓ |
| VGG-16 (Simonyan & Zisserman, 2015) | ✗ | 71.5 | 92.7 | 15.5 | 138.0 | R | ✗ |
| **Transformer-based** | | | | | | | |
| DeiT-S/16 (Touvron et al., 2021) | ✓ | 81.2 | - | 5.0 | 24.0 | G | ✓ |
| ViT-B/16 (Dosovitskiy et al., 2021) | ✗ | 77.9 | - | 55.5 | 86.0 | G | ✓ |
| **MLP-based** | | | | | | | |
| BiMLP-S (Xu et al., 2022) | ✗ | 70.0 | - | 1.21 | - | P | ✓ |
| BiMLP-B (Xu et al., 2022) | ✗ | 72.7 | - | 1.21 | - | P | ✓ |
| ResMLP-12 (Touvron et al., 2022) | ✓ | 76.6 | - | 3.0 | 15.0 | G | ✗ |
| Hire-MLP-Tiny (Guo et al., 2022) | ✗ | 79.7 | - | 2.1 | 18.0 | G | ✓ |
| CycleMLP-T (Chen et al., 2022) | ✗ | 81.3 | - | 4.4 | 28.8 | G | ✓ |
| ResMLP-24 (Touvron et al., 2022) | ✓ | 79.4 | - | 6 | 30.0 | G | ✗ |
| MLP-Mixer-B/16 (Tolstikhin et al., 2021) | ✗ | 76.4 | - | 11.6 | 59.0 | G | ✗ |
| MLP-Mixer-L/16 (Tolstikhin et al., 2021) | ✗ | 71.8 | - | 44.6 | 207.0 | G | ✗ |
| $S^2$MLP-Wide (Yu et al., 2022b) | ✗ | 80.0 | 94.8 | 14.0 | 71.0 | G | ✗ |
| $S^2$MLP-Deep (Yu et al., 2022b) | ✗ | 80.7 | 95.4 | 10.5 | 51.0 | G | ✗ |
| FF (Melas-Kyriazi, 2021) | ✗ | 74.9 | - | 7.21 | 59.0 | G | ✗ |
| **Polynomial-based** | | | | | | | |
| Π-Nets (Chrysos et al., 2020) | ✗ | 65.2 | 85.9 | 1.9 | 12.3 | None | ✗ |
| Hybrid Π-Nets (Chrysos et al., 2020)† | ✗ | 70.7 | 89.5 | 1.9 | 11.9 | R,T | ✗ |
| PDC-comp (Chrysos et al., 2022)† | ✗ | 69.8 | 89.9 | 1.3 | 7.5 | R,T | ✗ |
| PDC (Chrysos et al., 2022)† | ✗ | 71.0 | 89.9 | 1.6 | 10.7 | R,T | ✗ |
| $\mathcal{R}$-PolyNets (Chrysos et al., 2023) | ✗ | 70.2 | 89.3 | 1.9 | 12.3 | None | ✗ |
| $\mathcal{D}$-PolyNets (Chrysos et al., 2023) | ✗ | 70.0 | 89.4 | 1.9 | 11.3 | None | ✗ |
| **Multi-stage MONet-T-LayerNorm (Ours)** | ✗ | **77.0** | **93.4** | 2.8 | 10.3 | None | ✗ |
| **Multi-stage MONet-T-BatchNorm (Ours)** | ✗ | **77.7** | **93.8** | 2.8 | 10.3 | None | ✗ |
| **Multi-stage MONet-S-LayerNorm (Ours)** | ✗ | **81.3** | **95.5** | 6.8 | 32.9 | None | ✗ |

Table 3: Experimental validation of different polynomial architectures on smaller datasets. The best are marked in **bold**. $\mathcal{D}$-PolyNets perform favorably to most of the other baseline polynomial networks and the rest baselines. MONet outperforms all the baselines and all polynomial networks in two of the four datasets, and shares the best accuracy with $\mathcal{R}$-PolyNets in another dataset.

| Model | CIFAR10 | SVHN | Oxford Flower | Tiny Imagenet |
|---|---|---|---|---|
| Resnet18 | 94.4 | 97.3 | 87.8 | 61.5 |
| MLP-Mixer | 90.6 | 96.8 | 80.2 | 45.6 |
| Res-MLP | 92.3 | 97.1 | 83.2 | 58.9 |
| $S^2$MLP-Deep-S | 92.6 | 96.7 | 93.0 | 59.3 |
| $S^2$MLP-Wide-S | 92.0 | 96.6 | 91.5 | 52.7 |
| Hybrid Pi-Nets | 94.4 | - | 88.9 | 61.1 |
| Π-Nets | 90.7 | 96.1 | 82.6 | 50.2 |
| PDC | 90.9 | 96.0 | 88.5 | 45.2 |
| $\mathcal{R}$-PolyNets | 94.5 | 97.6 | 94.9 | 61.5 |
| $\mathcal{D}$-PolyNets | 94.7 | 97.5 | 94.1 | **61.8** |
| MONet-T | **94.8** | **97.6** | **95.0** | 61.5 |

contrary, the proposed model can recover the equations behind the dynamic system and explicitly restore the symbolic representation.

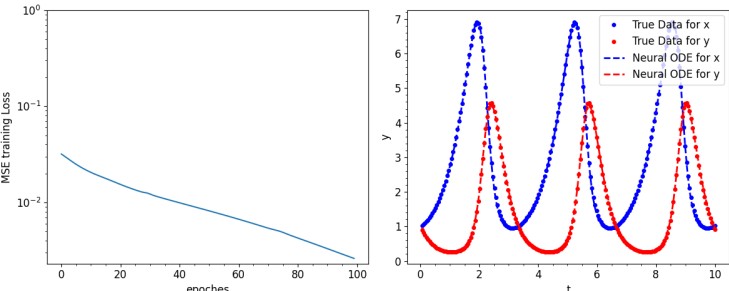

Figure 2: The training loss change with epochs trained(**Left**). The ground truth and model predicted trajectory. (**Right**) Our model achieves low loss in 20 epochs and successfully predicts real trajectory.

Table 4: Robustness on ImageNet-C (Hendrycks & Dietterich, 2019). The corruption error (and mean corruption error (mCE)) is used as the robustness metric, where a lower score indicates a better performance. The best performance per category is highlighted in bold. Notice that the proposed method is robust to a range of corruptions. In fact, in many categories, such as 'Weather' and 'Digital', the proposed model outperforms the compared models in all designated corruptions.

| | | Noise | | | Blur | | | | Weather | | | | Digital | | | |
|---|---|---|---|---|---|---|---|---|---|---|---|---|---|---|---|---|
| Network | mCE($\downarrow$) | Gauss | Shot | Impulse | Defocus | Glass | Motion | Zoom | Snow | Frost | Fog | Bright | Contrast | Elastic | Pixel | JPEG |
| ResNet50 | 76.7 | 79.8 | 81.6 | 82.6 | 74.7 | 88.6 | 78 | 79.9 | 77.8 | 74.8 | 66.1 | 56.6 | 71.4 | 84.7 | 76.9 | 76.8 |
| DeiT | 54.6 | 46.3 | 47.7 | 46.4 | 61.6 | **71.9** | 57.9 | 71.9 | 49.9 | 46.2 | 46 | 44.9 | 42.3 | 66.6 | 59.1 | 60.4 |
| Swin | 62.0 | 52.2 | 53.7 | 53.6 | 67.9 | 78.6 | 64.1 | 75.3 | 55.8 | 52.8 | 51.3 | 48.1 | 45.1 | 75.7 | 76.3 | 79.1 |
| MLP-Mixer | 78.8 | 80.9 | 82.6 | 84.2 | 86.9 | 92.1 | 79.1 | 93.6 | 78.3 | 67.4 | 64.6 | 59.5 | 57.1 | 90.5 | 72.7 | 92.2 |
| ResMLP | 66.0 | 57.6 | 58.2 | 57.8 | 72.6 | 83.2 | 67.9 | 76.5 | 61.4 | 57.8 | 63.8 | 53.9 | 52.1 | 78.3 | 72.9 | 75.3 |
| gMLP | 64 | 52.1 | 53.2 | 52.5 | 73.1 | 77.6 | 64.6 | 79.9 | 77.7 | 78.8 | 54.3 | 55.3 | 43.6 | 70.6 | 58.6 | 67.5 |
| CycleMLP | 53.7 | **42.1** | **43.4** | **43.2** | 61.5 | 76.7 | 56.0 | 66.4 | 51.5 | 47.2 | 50.8 | 41.2 | 39.5 | 72.3 | 57.5 | 56.1 |
| HireMLP | 51.9 | 52.4 | 55.3 | 55.3 | 60.3 | **71.6** | 59.6 | 57.8 | 54.3 | 52.5 | 43.5 | 29.1 | 42.2 | 54.9 | 49.7 | 40.2 |
| $\mathcal{R}$-PolyNets | 73.8 | 84.0 | 84.4 | 88.0 | 81.9 | 83.6 | 75.0 | 77.9 | 71.8 | 72.2 | 69.4 | 45.7 | 67.4 | 65.8 | 74.8 | 65.4 |
| MONet-S | **49.7** | 51.3 | 52.3 | 53.0 | **57.8** | 72.1 | **55.0** | **60.6** | **50.4** | **46.0** | **42.0** | **27.6** | **38.1** | **53.7** | **47.7** | **39.0** |

We provide an experiment of a polynomial neural ODE approximating the Lotka-Volterra ODE model. This model captures the dynamics of predator-prey population within a biological system. The equations representing the Lotka-Volterra formula are in the form of $\frac{dx}{dt} = \alpha x - \beta xy$, $\frac{dy}{dt} = -\delta y + \gamma xy$. Given $\alpha = 1.56, \beta = 1.12, \delta = 3.10, \gamma = 1.21$, the ground truth equation are presented below:

$$\frac{dx}{dt} = 1.56x - 1.12xy \, , \qquad\qquad \frac{dy}{dt} = -3.10y + 1.21xy \, . \qquad (2)$$

We generate $N = 100$ discrete points between time 0 and 10 for training. The training loss with epochs and predicted trajectory are shown in Fig. 2.

Notice that our model can directly recover the right hand side of the Lotka-Volterra formula. Indeed, the learned model in our case recovers the following formula:

$$\frac{dx}{dt} = 1.56x - 1.12001xy \, , \qquad\qquad \frac{dy}{dt} = -3.10y + 1.21001xy \, . \qquad (3)$$

Examining the results, it is evident that our Poly Neural ODE achieves high accuracy in recovering the ground truth formula and demonstrates rapid convergence. A more comprehensive comparison with NeuralODE and Augmented NeuralODE is conducted in Appendix G.

## 4.4 ROBUSTNESS

We further conduct experiments on ImageNet-C (Hendrycks & Dietterich, 2019) to analyze the robustness of our model. ImageNet-C contains 75 types of corruptions, which makes it a good testbed for our evaluation. We follow the setting of CycleMLP (Chen et al., 2022) and compare MONet with existing models using the corruption error as the metric (lower value indicates a better performance). Table 4 illustrates that our models exhibits the strongest ability among recent models. Notice that in important categories, such as the 'weather' or the 'digital' category, MONet outperforms the

Table 5: The influence of the different module in Block. Based on depth 16, hidden size 384 model on ImageNet100. We incorporate spatial shift since this operation is multilinear and brings a performance gain. However, the primary performance improvement comes from the Mu-Layer we proposed as shown in the following table. Notice all models mentioned below are without activation functions.

| Block Type | Layer 1 | Layer 2 | Paras. (M) | Top-1 Acc(%) |
|---|---|---|---|---|
| Poly-Block | Mu-Layer+Spatial-Shift | Mu-Layer | 1.70 | 82.94% |
| Poly-Block* | Mu-Layer | Mu-Layer | 1.70 | 81.50% |
| Mix Block | Mu-Layer+Spatial-Shift | MLP Layer | 1.26 | 67.61% |
| Linear Block | MLP Layer+Spatial-Shift | MLP Layer | 1.21 | 55.11% |

Table 6: The influence of the patch embedding approach on a model with depth 16, hidden size 384 trained on ImageNet100. Notice that the pyramid patch embedding can improve the performance with a minor difference in the parameters and maintain small FLOPs when using small patch.

| Method | Patch size | Top-1 Acc(%) | Paras (M) | FLOPs |
|---|---|---|---|---|
| Patch Embedding | 7 | 83.08 | 27.77 | 28.14 |
| Patch Embedding | 14 | 79.54 | 27.94 | 7.07 |
| Pyramid Patch Embedding | 7 | 82.04 | 28.02 | 7.23 |
| Pyramid Patch Embedding | 14,7 | 82.94 | 28.54 | 7.28 |

compared methods in all types of corruption. Those categories can indeed be corruptions met in realistic scenaria, making us more confident on the robustness of the proposed model in such cases.

## 4.5 ABLATION STUDY

We conduct an ablation study using ImageNet100, a subset of ImageNet1K with 100 classes selected from the original ones. Previous studies have reported that ImageNet100 serves as a representative subset of ImageNet1K (Yan et al., 2021; Yu et al., 2022a; Douillard et al., 2022). Therefore, ImageNet100 enables us to make efficient use of computational resources while still obtaining representative results for the self-evaluation of the model. In every experiment below, we assess one module or hyperparameter, while we assume that the rest remain the same as in the final model.

**Module ablation**: In this study, we investigate the influence of various modules on the final network performance by removing and reassembling modules within the block. Please note that the following blocks do not contain any activation functions. In Table 5, we report the result using a hidden size of 384 and depth 16 on the ImageNet100. We observe that spatial shift contributes to the performance improvement in our model but it is not crucial. The network still achieves high accuracy even in the absence of this module. At the same time, the removal of the Mu-Layer from the model leads to a significant drop in performance. The results validate that the spatial shift module alone cannot replace the proposed multilinear operations.

**Patch Embedding**: In this study, we validate the effectiveness of our proposed pyramid patch embedding. We replace the input embedding layer of a model with depth 16 and hidden size 384 on the ImageNet100 dataset. As depicted in Table 6, the model utilizing multi-level embedding achieves an Multi-stage performance with a Top-1 accuracy of 82.94%. The model utilizing a single-level embedding achieves a Top-1 accuracy of 81.78%, which is slightly lower than the multi-level embedding approach. For normal patch embedding, the computational and parameter overhead increases quadratically with a decrease in patch size. The result highlights that our approach significantly improves performance while adding only a small number of parameters and computational overhead. We conduct the same study for MLP-Mixer in Appendix D with similar outcomes, indicating the efficacy of our core Poly-Block independently of the patch embedding module.

**Hidden size**: We evaluate the impact of different hidden size $m$. We fix the depth of the network to 16 and vary the hidden size. We observe a sharp decrease when MLP-Mixer uses a small hidden size under 128. The results in Table 7 validate that our proposed method is more robust in hidden size change compared to the normal MLP layer.

Table 7: The influence of the hidden size, $m$. The models result are based on depth 16 model with different hidden size, the number in the second and their row indicates Top-1 Accuracy(%) on ImageNet100. The MLP-Mixer shows a sharp decline when hidden size change from 128 to 96.

| $m$ | 96 | 128 | 192 | 256 | 384 | 512 |
|---|---|---|---|---|---|---|
| MLP-Mixer | 48.3% | 70.05% | 72.39% | 76.88% | 78.24% | 77.93% |
| MONet | 67.5% | 75.82% | 79.3% | 81.94% | 82.94% | 82.24% |

Table 8: The influence of the depth, $N$. The model is based on depth 16, hidden size 384 and is trained on ImageNet100.

| $N$ | Top-1(%) | Top-5(%) | Paras (M) |
|---|---|---|---|
| 1 | 44.25 | 71.18 | 2.96 |
| 3 | 70.58 | 78.30 | 6.37 |
| 6 | 78.30 | 93.20 | 11.49 |
| 12 | 80.03 | 94.12 | 21.72 |
| 16 | 82.94 | 95.04 | 28.54 |

Table 9: The influence of the shrinkage ratio, $r$. The model is based on depth 16, hidden size $m = 384$, trained on ImageNet100.

| $r$ | Top-1(%) | Top-5(%) | Paras (M) |
|---|---|---|---|
| 1 | 83.18 | 94.86 | 52.98 |
| 2 | 83.88 | 95.54 | 36.46 |
| 4 | 82.94 | 95.04 | 28.54 |
| 8 | 82.28 | 95.12 | 24.06 |
| 16 | 82.34 | 94.62 | 21.99 |

**Depth**: The backbone of our proposed model, MONet, consists of $N$ blocks. We evaluate the influence of the number of blocks (depth $N$) on Top-1 accuracy and parameter numbers, using a model with a fixed hidden size and shrinkage ratio. We observe that for our network, the depth holds a more significance role than the hidden size. Our experiments result in Table 8 validate our theoretical conjecture, as increasing depth yields a more pronounced improvement in performance.

**Shrinkage ratio**: As mentioned in Section 3.1 the shrinkage ratio is defined as $r = \frac{m}{l}$. The results of different shrinkage ratios based on a hidden size of $m = 384$ and a depth of 16 in our model are presented in Table 9. We observe that: i) A larger shrinkage ratio effectively reduces the number of parameters while having a relatively minor impact on performance; ii) a lower effective rank in the weights is sufficient for performance gain.

## 5 CONCLUSION

In this work, we introduce a model, called MONet that expresses the output as a polynomial expansion of the input elements. MONet leverages *solely* linear and multilinear operations, which avoids the requirement for activation functions. At the core of our model lies the Mu-Layer, which captures multiplicative interactions inside the token elements (i.e. input). Through a comprehensive evaluation, we demonstrate that MONet surpasses recent polynomial networks, showcasing performance levels outperforms modern transformers models across a range of challenging benchmarks in image recognition. We anticipate that our work will further encourage the community to reconsider the role of activation functions and actively explore alternative classes of functions that do not require them. Lastly, we encourage the community to extend our illustration of the polynomial ODE solver in order to tackle scientific applications with PNs.

**Limitation**: A theoretical characterization of the polynomial expansions that can be expressed with MONet remains elusive. In our future work, we will conduct further theoretical analysis of our model. We believe that such an analysis would further shed light on the inductive bias of the block and its potential outside of image recognition.

## REPRODUCIBILITY STATEMENT

Throughout this study, we exclusively utilize publicly accessible benchmarks, guaranteeing that other researchers can replicate our experiments. Additionally, we provide comprehensive information about the hyperparameters employed in our study and strive to offer in-depth explanations of all the techniques employed. Our plan is to make the source code of our model open source once our work gets accepted.

## ACKNOWLEDGMENTS

We are thankful to Dr. Giorgos Bouritsas and the ICLR reviewers for their feedback and constructive comments. We are also thankful to Colby Fronk for his help in the NeuralODE symbolic representation restoration. We thank Zulip[7] for their project organization tool. This work was supported by the Hasler Foundation Program: Hasler Responsible AI (project number 21043). Research was sponsored by the Army Research Office and was accomplished under Grant Number W911NF-24-1-0048. This work was supported by the Swiss National Science Foundation (SNSF) under grant number 200021_205011.

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

## A PROOFS

In this section, we derive the proof of Prop. 1 and also include a new proposition for the Poly-Block, along with the associated proof.

### A.1 PROOF OF PROP. 1

In this paper, we use a bold capital letter to represent a matrix, e.g., $\boldsymbol{X}, \boldsymbol{F}$, while a plain letter to represent a scalar. For instance, $X_{j,\rho}$ represents the element in the $j^{\text{th}}$ row and the $\rho^{\text{th}}$ column. The matrix multiplication of two matrices $\boldsymbol{F}, \boldsymbol{K}$ results in the following $j, \rho$ element: $[\boldsymbol{F} \cdot \boldsymbol{K}]_{j,\rho} = \sum_q F_{j,q} K_{q,\rho}$.

As a reminder, the input is $\boldsymbol{X} \in \mathbb{R}^{d \times s}$, where $d \in \mathbb{N}$ is the length of a token and $s \in \mathbb{N}$ is the number of tokens. Thus, to prove the proposition, we need to show that products of the form $X_{\tau,\rho} X_{\psi,\rho}$ exist between elements of each token.

Eq. (1) relies on matrix multiplications and a Hadamard (elementwise) product. Concretely, if we express the $j, \rho$ element of $(\boldsymbol{AX}) * (\boldsymbol{BDX})$, we obtain:

$$[(\boldsymbol{AX}) * (\boldsymbol{BDX})]_{j,\rho} = \sum_{\tau=1,\psi=1}^{d} \sum_{\omega} A_{j,\tau} X_{\tau,\rho} B_{j,\omega} D_{\omega,\psi} X_{\psi,\rho} . \tag{4}$$

Then, we can add the additive term and the matrix $\boldsymbol{C}$. If we express Eq. (1) elementwise, we obtain the following expression:

$$[\boldsymbol{Y}]_{i,\rho} = \sum_{j=1}^{m} C_{i,j} \left\{ \sum_{\tau=1,\psi=1}^{d} \sum_{\omega} A_{j,\tau} X_{\tau,\rho} B_{j,\omega} D_{\omega,\psi} X_{\psi,\rho} + \sum_{\tau=1}^{d} A_{j,\tau} X_{\tau,\rho} \right\} . \tag{5}$$

That last expression indeed contains sums of products $X_{\tau,\rho} X_{\psi,\rho}$ for every output element, which concludes our proof.

### A.2 INTERACTIONS OF THE POLY-BLOCK

As we mention in the main paper, a Poly-Block comprises of two Mu-Layers, so one reasonable question is how the multiplicative interactions of the Mu-Layer extend in the Poly-Block. We prove below that up to fourth degree interactions are captured in such a block. To simplify the derivation, we focus on two consecutive Mu-Layers as the Poly-Block.

**Proposition 2.** *The Poly-Block captures up to fourth degree interactions between elements of each token.*

*Proof.* All we need to show is that products of the form $X_{\tau,\rho} X_{\psi,\rho} X_{\gamma,\rho} X_{\epsilon,\rho}$ appear. Using the insights from Appendix A.1, we expect the fourth degree interactions to appear in the Hadamard product, so we will focus on the term $(\boldsymbol{A}^{(2)} \boldsymbol{X}^{(2)}) * (\boldsymbol{B}^{(2)} \boldsymbol{D}^{(2)} \boldsymbol{X}^{(2)})$, where the (2) declares the weights and input of the second Mu-Layer.

The elementwise expression for $\boldsymbol{X}^{(2)}$ is directly obtained from Eq. (5). To simplify the expression, we ignore the additive term, since exhibiting the fourth degree interactions is enough. The $q, w$ element of the expression is the following:

$$\left[ \left( \boldsymbol{A}^{(2)} \boldsymbol{X}^{(2)} \right) * \left( \boldsymbol{B}^{(2)} \boldsymbol{D}^{(2)} \boldsymbol{X}^{(2)} \right) \right]_{w,\rho} = \sum_{q,\delta} \sum_{\xi=1,\theta=1}^{m} \sum_{\alpha} C_{q,\xi} C_{\delta,\theta} A_{w,q}^{(2)} B_{w,\alpha}^{(2)} D_{\alpha,\delta}^{(2)} \{f_1\} \{f_2\} , \tag{6}$$

where the expression $\{f_1\} \{f_2\}$ is the following:

$$\sum_{\tau=1,\psi=1,\gamma=1,\epsilon=1}^{d} \sum_{\omega,\omega^\dagger} A_{\xi,\tau} X_{\tau,\rho} B_{\xi,\omega} D_{\omega,\psi} X_{\psi,\rho} A_{\theta,\gamma} X_{\gamma,\rho} B_{\theta,\omega^\dagger} D_{\omega^\dagger,\epsilon} X_{\epsilon,\rho} . \tag{7}$$

Notice that the last expression indeed contains products of the form $X_{\tau,\rho} X_{\psi,\rho} X_{\gamma,\rho} X_{\epsilon,\rho}$, which concludes the proof. □

# B  MULTI-STAGE MONET

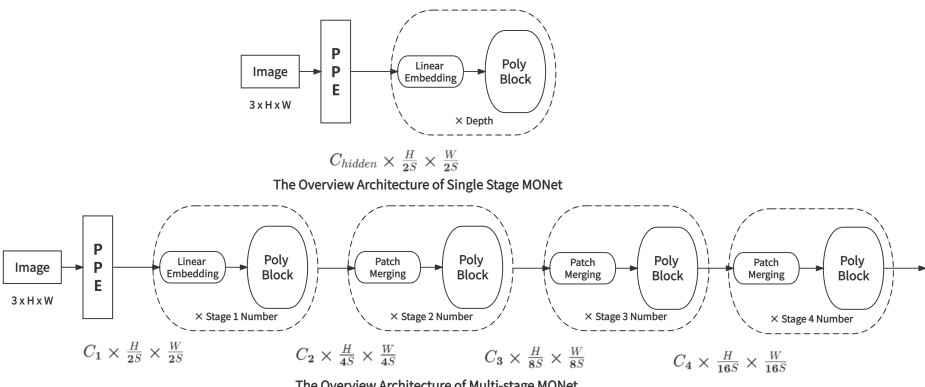

Figure 3: The Schematic of (simple) MONet and Multi-stage MONet. PPE represents our pyramid patch embedding.

In deep architectures often a different number of channels or hidden size is used for different blocks (He et al., 2016). Our preliminary experiments indicate that MONet is amenable to different hidden size as well. Following recent works (Chen et al., 2022; Guo et al., 2022), we set 4 different hidden sizes across the network (referred to as stages) and we refer to this variant as the 'Multi-stage MONet'. This variant is mostly used for large-scale experiments, since for datasets such as CIFAR10, and CIFAR100 a single hidden size is sufficient.

# C  DETAILS OF MU-LAYER AND POLY-BLOCK

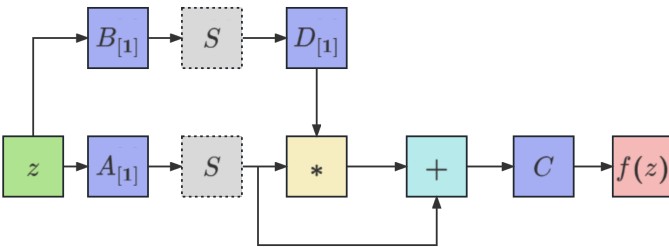

Figure 4: The Schematic of Mu-Layer. Blue boxes correspond to learnable parameters. Green and red boxes denote input and output, respectively. The $*$ denotes the Hadamard product, the $+$ denotes element-wise addition. The gray box denotes the spatial aggregation module, the dotted line represents it as an optional module. In our design the first Mu-Layer of each Poly-Block includes a spatial aggregation unit, while the second Mu-Layer does not.

In Fig. 4, we present the schematic of the Mu-Layer. The matrices $A$, $B$, $C$, and $D$ in Fig. 4 are described in Section 3.1. As mentioned in Section 3.1, the Poly-Block serves as a larger unit module in our proposed model. In our implementation, the Poly-Block consists of two Mu-Layer modules. Only the first Mu-Layer in each block incorporates a spatial aggregation module, where we utilize a spatial shift operation (Yu et al., 2022b).

# D  PYRAMID EMBEDDING

A Patch Embedding, which is typically used in the input space of the model, converts an image into a sequence of $N$ non-overlapping patches and projects to dimension $d$. If the original image has

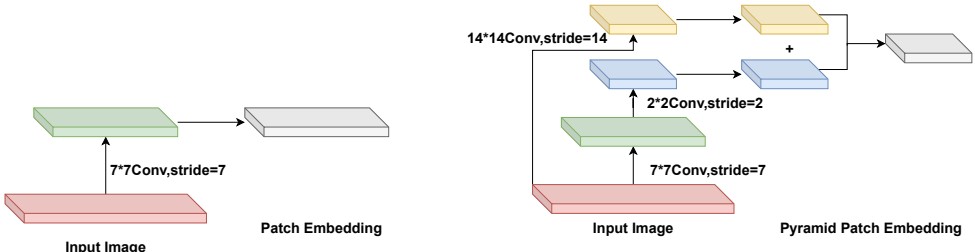

Figure 5: Pyramid Patch Embedding

Table 10: Influence of the patch embedding on MONet.

| Method | Patch size | Top-1 Acc(%) | Paras (M) | FLOPs |
|---|---|---|---|---|
| Patch Embedding | 7 | 83.08 | 27.77 | 28.14 |
| Patch Embedding | 14 | 79.54 | 27.94 | 7.07 |
| Pyramid Patch Embedding | 7 | 82.04 | 28.02 | 7.23 |
| Pyramid Patch Embedding | 14,7 | 82.94 | 28.54 | 7.28 |

Table 11: The influence of the patch embedding approach on MLP-Mixer.

| Method | Patch size | Top-1 Acc(%) | Paras (M) | FLOPs |
|---|---|---|---|---|
| Patch Embedding | 7 | 77.92 | 26.60 | 19.44 |
| Patch Embedding | 14 | 77.93 | 23.68 | 4.91 |
| Pyramid Patch Embedding | 7 | 79.64 | 24.50 | 5.18 |
| Pyramid Patch Embedding | 14,7 | 79.96 | 24.80 | 5.26 |

$(H, W)$ resolution, each patch $X_i$ has resolution $(P, P)$. Then, we obtain $S = H \times W/P^2$ patches, where $X_i \in \mathbb{R}^{P \times P \times d}$. For models based on MLP structure, smaller patches can capture finer-grained features better. However, due to the quadratic growth in computation caused by reducing the patch size, most models (Guo et al., 2022; Chen et al., 2022) do not follow this design. To reduce the computational burden by smaller patches, these networks assume multiple hidden sizes across the network, see Appendix B for further details. Instead, we utilize an additional method for reducing the computational cost. After performing the first patch embedding on the original image using a small patch size, we can further reduce the input size by performing $2 \times 2$ convolution with stride 2. With this approach, we can achieve performance comparable to directly using a small patch size while significantly reducing computational costs and allowing us to adopt a more elegant and simple structure.

Inspired by FPN (Lin et al., 2017), we introduce a different level of patch embedding to further enhance the performance. Each level of the pyramid represents an image at a different scale, with the lower levels representing images with higher resolution and the higher levels representing images with lower resolution. We use lower resolution image patch merged with downsampled higher-resolution image patch by element-wise addition. In Section 4.5, we compare our method with normal patch embedding. The embedding method we propose not only reduces network computational complexity but also improves the performance when compared to larger patch sizes.

We conduct 8 comparisons to prove the soundness of our method. We employ different patch embedding and patch sizes for the MLP-Mixer and our model. We train those models on ImageNet-100 dataset and report Top-1 Accuracy, their parameters and computation costs (FLOPs) in Table 10 and Table 11.

Appendix D indicates that our pyramid patch embedding improves the performance compared to normal patch embedding. By leveraging our proposed method, we can utilize small patch size to achieve better results while preventing quadratic growth of computation costs. This could be used as an ad-hoc module in other MLP models. At the same time, even with a normal patch embedding, our model still outperforms MLP-Mixer.

In the context of merging different levels of features, we initially adopt an elementwise addition approach. Additionally, we explore a U-Net-style approach (Ronneberger et al., 2015). In this approach, we concatenate the features along the channel dimension and then reduced the dimension using a 1x1 depthwise convolution. However, after conducting experiments, we observe that this approach had limited improvement and was considerably less effective compared to the elementwise addition method.

Table 12: MONet performance compared to ResNet18 on the MedMNIST benchmark, with detailed numbers. The best results are marked in **bold**.

| Dataset | MONet-T | PDC | Π-Nets | MLP-Mixer | ResMLP | $S^2$MLP-D-S | $S^2$MLP-W-S | ResNet | AutoSklearn |
|---------|---------|-----|--------|-----------|--------|--------------|--------------|--------|-------------|
| Path | **90.8** | 92.1 | 90.8 | 89.2 | 89.6 | 90.1 | 90.7 | 90.7 | 83.4 |
| Derma | **77.5** | 77.5 | 72.9 | 76.1 | 76.5 | 76.2 | 76.9 | 73.5 | 74.9 |
| Oct | **80.1** | 77.2 | 79.8 | 78.2 | 79.7 | 79.1 | 79.5 | 74.3 | 60.1 |
| Pneumonia | **93.4** | 92.5 | 89.1 | 93.1 | 89.1 | 91.9 | 93.1 | 85.4 | 87.8 |
| Retina | **55.7** | 53.6 | 52.2 | 54.5 | 54.7 | 53.7 | 54.0 | 52.4 | 51.5 |
| Blood | **96.7** | 95.5 | 94.5 | 94.7 | 95.3 | 95.8 | 95.2 | 93.0 | 96.1 |
| Tissue | **67.7** | 67.5 | - | 65.9 | 66.8 | 65.9 | 66.0 | 67.7 | 53.2 |
| OrganA | **93.6** | 93.5 | 92.5 | 90.5 | 91.4 | 92.3 | 92.7 | 93.5 | 76.2 |
| OrganC | 88.9 | 93.0 | 89.3 | **90.6** | 87.5 | 88.9 | 89.4 | 90.0 | 87.9 |
| OrganS | **78.5** | 77.9 | 75.0 | 76.9 | 75.7 | 77.5 | 78.4 | 78.2 | 67.2 |

## E  MEDICAL IMAGE CLASSIFICATION

To assess the efficacy of our model beyond natural images, we conduct an experiment on the MedM-NIST challenge across ten datasets (Yang et al., 2021). The dataset encompasses diverse medical domains, such as chest X-rays, brain MRI scans, retinal fundus images, and more. MedMNIST serves as a benchmark dataset for evaluating models in the field of medical image analysis, enabling us to evaluate the performance of our model across various medical imaging domains.

In our experiments, we train the variant of MONet-T with 14.0M parameters and 0.68G FLOPs. The results are shown in Table 12 exhibit that our model outperforms other models in eight datasets.

## F  VISUALIZATION OF REPRESENTATIONS FROM LEARNED MODELS

In this section, we use visualizations to understand the difference between the learned models of MLP-Mixer, ResMLP and CNN architectures. The effective receptive field refers to the region in the input data that a neural network's output is influenced by, which has been proved as an effective approach to understand where a learned model focuses on. Specifically, we visualize the hidden unit effective receptive field of our model trained on ImageNet1k, i.e., the output of the last layer, and compare it with pretrained MLP-Mixer, ResMLP and Resnet in Fig. 6. We adopt a random image from ImageNet1K as input and visualize the effective receptive field. We can notably observe that due to the flattening of input tokens into one-dimensional structures by MLP-Mixer and ResMLP, the features they learn exhibit a distinct grid effect due to the loss of 2D information. They also tend to emphasize low-level texture information to a greater extent. ResNet's effective receptive field is more discrete, encompassing both background and foreground elements to some extent, focusing on the global context. On the contrary, MONet focuses on the semantic parts (e.g. on the dog's face) which achieves a balance between global and local context.

## G  NEURALODE SOLVER COMPARSION

Following the evaluation setup of the Neural-ODE network, we implement a Poly-Neural ODE based on our method, which can be used to solve ordinary differential equations for simulating physics problems. We conduct scientific computing experiments using simulated data, with the aim of predicting and interpreting the dynamics of physical systems

The specific task is to simulate moving the **randomly generated** inner sphere particles out of the annulus by pushing against and stretching the barrier. Eventually, since we are mapping discrete points and not a continuous region, the flow is able to break apart the annulus to let the flow through. The initial particle states are shown in Figure Fig. 7.

We set the model with hidden dimension 32 and train on simulated data for 12 epoches. The revolution of the first 10 epochs are shown in Fig. 8.

We implement two models based on NeuralODE (Chen et al., 2018a) and Augmented Nerual ODE (Dupont et al., 2019), called Poly NerualODE(PolyNODE) and Poly Augmented NeuralODE(Poly

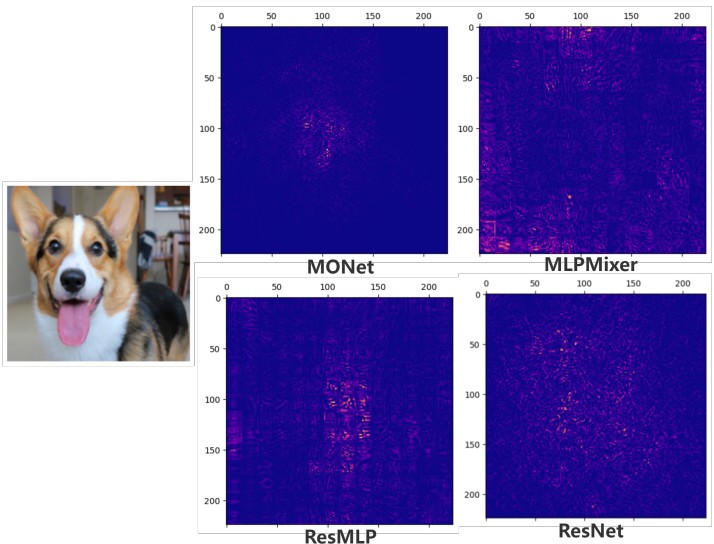

Figure 6: Input image(**Left**). The ERF of model pretrained on ImageNet-1K(**Right**). We visualize the effective receptive field of MLP-Mixer, ResMLP, ResNet and MONet. Our model more focus on the dog's face and the outline shape of dog, while the rest MLP models more focus on the low-level texture feature.

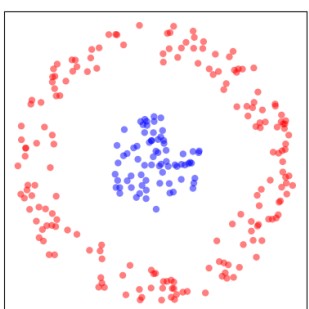

Figure 7: The initial particles states. The task is to move the inner sphere of blue particles out of the red particles.

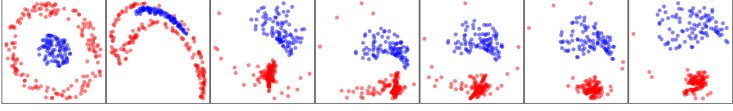

Figure 8: The move of particles. From left to right it's feature evolution with iterations increase.

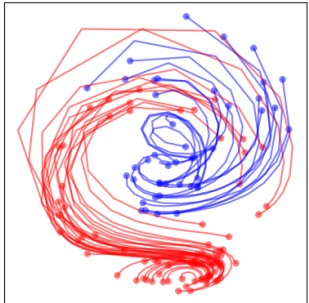

Figure 9: The predicted trajectory of particles. The lines are their history trajectory.

ANODE). For Poly NeuralODE, higher-order model could achieve better accuracy with the cost of higher computation cost, the following experiment are based on minimal order 2 Poly NeuralODE.

We first compare NODE with Poly NODE, the loss plots shown in Fig. 10. We can observe that with 40 epochs training, The Poly NODE approximates the functions faster and more accurately. We then

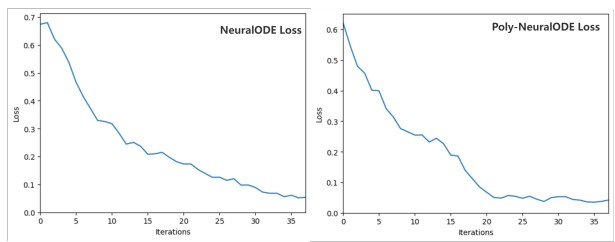

Figure 10: Loss plots for NODEs and ANODEs trained on 2d data. Poly NODEs easily approximate the functions and are consistently faster than NODEs.

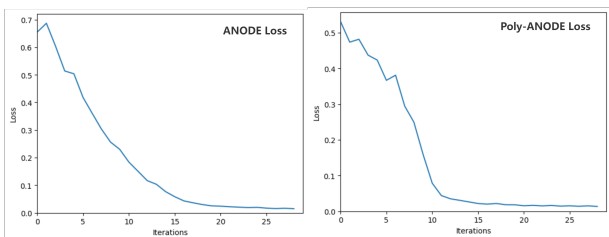

Figure 11: Loss plots for Poly ANODEs and ANODEs trained on 2d data. Poly NODEs easily approximate the functions and are consistently faster than NODEs.

compare ANODE with Poly ANODE. The loss plot trained 30 epoches is shown in Fig. 11. ANODEs augment the space on which the ODE is solved, allowing the model to use the additional dimensions to learn more complex functions using simpler flows. Our PolyANODE inherent its advantage while converge faster.

## H SEMANTIC SEGMENTATION

**Settings.** To further explore the performance of MONet in downstream task. We conduct semantic segmentation experiments on ADE20K dataset and present the final performance compared to previous models. The Appendix H shows the result. Previous PNN models exhibited poor performance on downstream tasks. Our model has overcome this issue and achieved results comparable to some state-of-the-art models.

## I COMPUTATION COST COMPARED TO PREVIOUS POLYNOMIAL MODELS

FLOPs (Floating-Point Operations Per Second) is commonly used metric to measure the computational cost of a model. However, tools like flops-counters and mmcv, which use PyTorch's module forward hooks, have significant limitations. They are accurate only if every custom module includes a corresponding flop counter.

In this paper, we adopt fvcore Flop Counter[8] developed by Meta FAIR, which provides the first operator-level FLOP counters. This tool observes all operator calls and collects operator-level FLOP counts. For models like polynomial models that involve numerous custom operations, an operator-level counter will give more accurate results. We reproduced previous polynomial models according

---

[8]fvcore flops counter

Table 13: **Semantic segmentation on ADE20K val**,All models use Semantic FPN head. The transformer and ResNet models result from the original paper. The polynomial networks result from the reproduced experiment.

| Backbone | Semantic FPN |
|---|---|
| | mIoU(%) |
| ResNet18 (He et al., 2016) | 32.9 |
| FCN (Long et al., 2015) | 29.3 |
| PvT-Tiny (Wang et al., 2021) | 35.7 |
| Seg-Former (Xie et al., 2021) | 37.4 |
| R-PDC (Chrysos et al., 2022) | 20.7 |
| $\mathcal{R}$-PolyNets(Chrysos et al., 2023) | 19.4 |
| Multi-stage MONet-S | 37.5 |

to their open-source code and re-measure their computation cost with fvcore flops counter. This causes a slight difference from the FLOPs reported in previous PNs. For instance, operations such as custom normalization modules and Hadamard product computations were often overlooked due to former tool limitations. *We adopt the number taken from original papers in Table 2 and we have corrected and updated the FLOPs report for recent polynomial models in the table below.* In comparison to previous work, our model has significant performance benefits while *only incurring 20% of the computational cost* of the previous state-of-the-art work. Simultaneously, our performance on ImageNet-1K surpasses the previous state-of-the-art by **10%**.

Table 14: The re-measure computational costs for polynomial models, along with their corresponding Top-1 accuracy on ImageNet-1K

| Model | FLOPs(B) | Accuracy(%) |
|---|---|---|
| PDC | 30.78 | 71.0 |
| R-PDC | 39.56 | - |
| $\mathcal{R}$-PolyNets | 29.34 | 70.2 |
| $\mathcal{D}$-PolyNets | 26.41 | 70.0 |
| MONet-T (Ours) | 3.6 | 76.0 |
| MONet-S(Ours) | 11.2 | 79.6 |
| Multi-stage MONet-T(Ours) | 2.8 | 77.0 |
| Multi-stage MONet-S(Ours) | 6.8 | 81.3 |

## J   COMPLEXITY ANALYSIS

To simplify the process, we conduct a complexity analysis of MONet adopted the single-stage design as follows.

**Pyramid Patch Embedding Layer (PPEL)** crops the input raw image $I \in \mathbb{R}^{H \times W \times 3}$ where $W$ is the width and $H$ is the height into several non-overlapping patches of dimension $P \in \mathbb{R}^{p \times p \times 3}$. Consider 1-level embedding,Our method conducts compression to the patches with a $2 \times 2$ convolution layer, resulting in the token $W_0 \in \mathbb{R}^{p \times p \times c}$, token $W_1 \in \mathbb{R}^{\frac{p}{2} \times \frac{p}{2} \times c}$,where $W_0$ is the output of the first convolution layer, $W_1$ is the output of the second convolution layer. Noted that the final number of patches are $N_p = \frac{W}{2p} \times \frac{H}{2p}$ which is 25% of normal patch embedding. Thus, the total number of parameters of the Pyramid embedding is:

$$Params_{PPEL} = (3p^2 + 1)c + c(4c + 1) = (3p^2 + 4c + 2)c \tag{8}$$

In summary the floating operations in PPEL is

$$FLOPs_{PPEL} = 3 \times \frac{W}{p} \times \frac{H}{p}cp^2 + 4 \times c^2 \times \frac{W}{2p} \times \frac{H}{2p} = 4cN_p(3p^2 + c) \tag{9}$$

**Poly Blocks(PB)**: Our model consists of $N$ identical Poly Blocks. Each blocks contains 2 Poly Layer. The first Poly Layers($PL_1$) contains two Spatial Shift module in two branches, and the

second Poly Layers doesn't contain Spatial Shift module. We denotes that first Poly Layer four fully connected layer as $W_1, b_1, W_2, b_2, W_3, b_3, W_4, b_4$ and shrinkage ratio as $s$, where $W_1 \in \mathbb{R}^{c \times c}, W_2 \in \mathbb{R}^{c \times \frac{c}{s}}, W_3 \in \mathbb{R}^{\frac{c}{s} \times c}, W_4 \in \mathbb{R}^{c \times c}$. Therefore, the total parameter numbers of the first Poly Layers is

$$Params_{PL_1} = c \times (2c + 2\frac{c}{s}) + 3c + \frac{c}{s} = c^2(2 + \frac{2}{s}) + c(3 + \frac{1}{s}) \qquad (10)$$

The second Poly Layer is similar, while we have a expansion rate $r$ similar to other MLP models. We denotes that second Poly Layer four fully connected layer as $W_5, b_5, W_6, b_6, W_7, b_7, W_8, b_8$ and shrinkage ratio as $s$, where $W_5 \in \mathbb{R}^{c \times rc}, W_2 \in \mathbb{R}^{c \times \frac{rc}{s}}, W_3 \in \mathbb{R}^{\frac{rc}{s} \times rc}, W_4 \in \mathbb{R}^{rc \times c}$. We get the total parameter numbers of the second Poly Layers is

$$Params_{PL_2} = c \times (2rc + \frac{rc}{s} + \frac{r^2c}{s}) + 2rc + c + \frac{rc}{s} = c^2(2r + \frac{r + r^2}{s}) + c(2r + \frac{1}{s} + 1) \quad (11)$$

Hence, the total number of a Poly Block is

$$Params_{PB} = Params_{PL_1} + Params_{PL_2} = c^2(2r + 2 + \frac{r^2 + r + 2}{s}) + c(4 + 2r + \frac{2}{s}) \quad (12)$$

Besides the fully connected layer, the hadamard product bring extra computation where the flops of hadamard product of two matrices in shape $n \times m$ is $n \times m$. The overall hadamard product flops is

$$FLOPs_{Prod} = N_p c + N_p rc = N_p c(1 + r) \qquad (13)$$

and the flops becomes

$$FLOPs_{PB} = N_p c^2(2r + 2 + \frac{r^2 + r + 2}{s}) + FLOPs_{Prod} = N_p c^2(2r + 2 + \frac{r^2 + r + 2}{s}) + N_p c(1 + r) \qquad (14)$$

**Fully-connected classification layer(FCL)**: takes input $c$-dimensional vector and feedforward to average pooling layer. The output vectore is in $k$ dimension where $k$ is the number of classes. In summary, the number of parameters of FCL is:

$$Params_{FCL} = k(c + 1) \qquad (15)$$

The flops of FCL is:

$$FLOPs_{FCL} = N_p ck \qquad (16)$$

**Overall architecture**: We conclude that the summary of parameters of overall of architecture. The total numbers of parameters of our architecture is:

$$Params = Params_{PPEL} + N \times Params_{PB} + Params_{FCL} \qquad (17)$$

The flops of our architecture is:

$$FLOPs = FLOPs_{PPEL} + N \times FLOPs_{PB} + FLOPs_{FCL} \qquad (18)$$

# K IMAGENET1K CLASSIFICATION

## K.1 IMAGENET1K TRAINING SETTING

The Appendix K.1 shows our experiment setting for training MONet on ImageNet1K dataset. Our implementation for the optimizer and data augmentation follow the popular timm library[9]. We train 300 epochs in ImageNet1K on 4 GPUs, we enable exponential moving average on 200th epochs.

Table 15: ImageNet1K Training Settings in Section 4.1

|  | ImageNet1K Training Setting |
| --- | --- |
| optimizer | AdamW |
| base learning rate | 1e-3 |
| weight decay | 0.01 (Multi-stage MONet-T) |
|  | 0.02 (Multi-stage MONet-S) |
| batch size | 448 |
| training epochs | 300 |
| learning rate schedule | cosine |
| warmup | ✓ |
| label smoothing | 0.1 |
| auto augmentation | ✓ |
| random erase | 0.1 |
| cutmix | 0.5 |
| mixup | 0.5 |

## K.2 ERROR ANALYSIS

We utilize our best-performing MONet model to compute per-class accuracy rates for all 1000 classes on the validation dataset of ImageNet1K. In Table 16, we present the top 10 least accurate and misclassified classes. Additionally, in Fig. 12 we showcase the images of the most misclassified class (laptop). In Fig. 13, we show a failure case of the screen. In Fig. 14, we show a failure case of tennis. In Fig. 15, we show a successful case of small object sunglasses.

Table 16: The top-10 least accurate classes and their labels

| Class Name | Accuracy (%) | Class Name | Accuracy (%) |
| --- | --- | --- | --- |
| tiger Cat | 20 | laptop,laptop computer | 24 |
| screen | 24 | chiffionier,commode | 28 |
| sunglasses | 28 | cassette player | 30 |
| letter opener | 30 | malliot | 32 |
| projectile,misslle | 32 | spotlight | 32 |

Our analysis dictates that the majority of classification failures occur when the model fails to focus on the main object. This is even more pronounced in cases where the classification label is provided by an item occupying a small portion of the image. We believe this is an interesting phenomenon that is shared with similar models relying on linear operations and not using convolutions.

---

[9]Hugging Face timm

Original Image         Saliency Map         Overlay

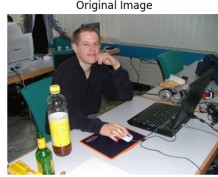 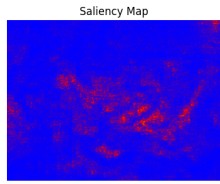 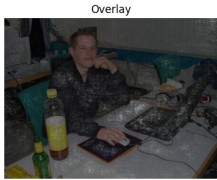

Figure 12: Failure case of MONet as explained from the saliency map. The red figure in the middle denotes the saliency map, which is the area where the model focuses on. The correct class for the image is laptop, while the saliency map shows the model focuses on the person. This is not an unreasonable error though, since the main figure lies in the center of the image.

Original Image         Saliency Map         Overlay

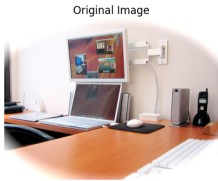 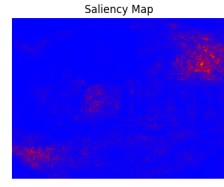 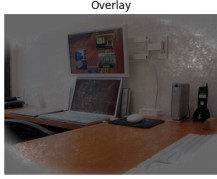

Figure 13: Failure case of MONet as explained from the saliency map. The red figure in the middle denotes the saliency map, which is the area where the model focuses on. The correct class for the image is screen, while the saliency map shows the model focusing on the wall.

Original Image         Saliency Map         Overlay

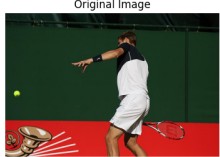 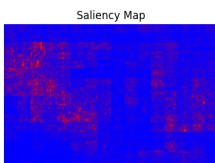 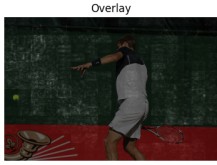

Figure 14: Failure case of MONet as explained from the saliency map. The red figure in the middle denotes the saliency map, which is the area where the model focuses on. The correct class for the image is tennis, while the saliency map shows the model focusing on the background wall, instead of on the small object.

Original Image         Saliency Map         Overlay

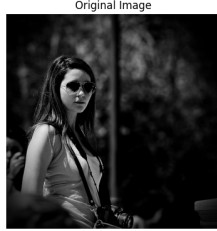 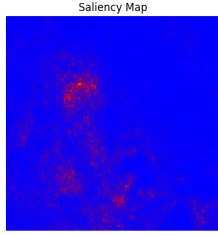 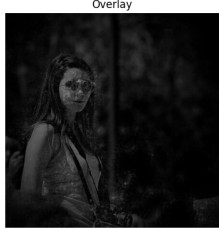

Figure 15: Success case of MONet as explained from the saliency map. The correct class for the image is sunglasses, The saliency map shows the model focusing on the face of the woman. In some cases, successful classification of a category can be achieved based on global information, even if the object occupies only a small portion of the image.

### K.3 Sources for baselines

To further increase the transparency of our comparisons, we clarify the details on the main results on ImageNet1K. We aim for reporting the accuracy and the other metrics using the respective papers or public implementations. Concretely:

- The paper of Chrysos et al. (2022) is used as the source for PDC.
- Chrysos et al. (2023) is used as the source for ResNet18,Pi-Nets, Hybrid Pi-Nets, $\mathcal{R}$-PolyNets and $\mathcal{D}$-PolyNets. We clarify here once more that Π-Nets refers to the model *without* activation functions, while 'Hybrid Π-Nets' refers to the model with activation functions.
- Yu et al. (2022b) is used as the source of ResNet50 performance. We updated the number to 77.2% (on ImageNet1K) to align with He et al. (2016).
- The performance data for the other models in the Table 2 are sourced from their respective papers.

## L    Additional benchmarks and Medical Image Classification

We introduce dataset details including dataset information for various datasets used for the empirical validation of our method.

In this section, we give a brief introduction to the dataset we use and the overview statistics of datasets are shown in Appendix L.

**CIFAR10**: CIFAR-10 (Krizhevsky et al., 2009) is a well-known dataset in the field of computer vision. It consists of 60,000 labeled images that are divided into ten different classes. Each image in the CIFAR-10 dataset has a resolution of 32x32 pixels and is categorized into one of the following classes: airplane, automobile, bird, cat, deer, dog, frog, horse, ship, or truck. The images in the CIFAR10 dataset are of size $32 \times 32$ pixels. We keep its original resolution for training.

**SVHN**:SVHN (Netzer et al., 2011) stands for Street View House Numbers. It is a widely used dataset for object recognition tasks in computer vision. The SVHN dataset consists of images of house numbers captured by Google's Street View vehicles. These images contain digit sequences that represent the house numbers displayed on buildings. It contains over 600,000 labeled images, which are split into three subsets: a training set with approximately 73,257 images, a validation set with around 26,032 images, and a test set containing approximately 130,884 images. The images in the SVHN dataset are of size $32 \times 32$ pixels, we keep its original resolution for training.

**Oxford Flower**: The Oxford Flower (Nilsback & Zisserman, 2008) dataset, also known as the Oxford 102 Flower dataset, is a widely used collection of images for fine-grained image classification tasks in computer vision. It consists of 102 different categories of flowers, with each category containing a varying number of images. The Oxford Flower dataset provides a challenging testbed for researchers and practitioners due to the high intra-class variability among different flower species. This variability arises from variations in petal colors, shapes, and overall appearances across different species. The images in the Oxford Flower dataset are of size $256 \times 256$ pixels, in our training, we use bicubic interpolation to resize all images to $224 \times 224$ pixels.

**Tiny-Imagenet**: Tiny ImageNet(Le & Yang, 2015) is a dataset derived from the larger ImageNet dataset, which is a popular benchmark for object recognition tasks in computer vision. The Tiny ImageNet dataset is a downsized version of ImageNet, specifically designed for research purposes and computational constraints. While the original ImageNet dataset consists of millions of images spanning thousands of categories, the Tiny ImageNet dataset contains 200 different classes, each having 500 training images and 50 validation and test images. This results in a total of 100,000 labeled images in the dataset. The images in the Tiny ImageNet dataset are of size $64 \times 64$ pixels, we keep its original resolution for training.

**MedMNIST**: MedMNIST (Yang et al., 2021) is a specialized dataset designed for medical image analysis and machine learning tasks. It is inspired by the popular MNIST dataset, but instead of handwritten digits, MedMNIST focuses on medical imaging data. The MedMNIST dataset includes several sub-datasets, each corresponding to a different medical imaging modality or task. Some

examples of these sub-datasets are ChestX-ray, Dermatology, OCT (Optical Coherence Tomography), and Retinal Fundus. Each sub-dataset contains labeled images that are typically 28x28 pixels in size, resembling the format of the original MNIST dataset. We keep the original size for training. Due to the presence of both the RGB sub-dataset and grayscale sub-dataset in MedMNIST, we employed different model configurations to accommodate this variation. The PathMNIST, DermaMNIST, BloodMNIST, and RetinaMNIST are RGB images, the rest 7 dataset are greyscale images.

**ImageNet-C**: ImageNet-C (Hendrycks & Dietterich, 2019) is a dataset of 75 common visual corruptions. This dataset serves as a benchmark for evaluating the resilience of machine learning models to different forms of visual noise and distortions. This benchmark provides a more comprehensive understanding of a model's performance and can lead to the development of more robust algorithms that perform well under a wider range of scenarios. The metric used in ImageNet-C is mean corruption error, which is the average error rate of 75 common visual corruptions of ImangeNet validation set. The lower the mCE, the better the result.

To ensure a fair comparison with the original paper that presented ResNet18, we followed the training protocol outlined in the MedMNIST paper. We trained our model for 100 epochs without any data augmentation, using early stopping as the stopping criterion.

Table 17: The overview of the dataset we use in Appendix E and Section 4.2. The * in the below table since MedMNIST dataset consists of multiple sub-datasets, each containing medical images from specific categories. These sub-datasets have varying numbers of training samples, testing samples, and classes.

| Dataset | CIFAR10 | SVHN | Oxford Flower | Tiny-Imagenet | MedMNIST |
|---|---|---|---|---|---|
| Classes | 10 | 10 | 102 | 200 | * |
| Train samples | 50000 | 73257 | 1020 | 100000 | * |
| Test samples | 10000 | 26032 | 6149 | 10000 | * |
| Resolution | 32 | 32 | 256 | 64 | 28 |
| Attribute | Natural Images | Numbers | Flowers | Natural Images | Medical Images |
| Image Type | RGB | RGB | RGB | RGB | RGB+Greyscale |

## M  INITIALIZATION

We evaluate various initialization methods for the parameters. The results in Table 18 indicate that the Xavier Normal initialization method yields the most favorable results. By adopting the Xavier Normal initialization, we observe improvements in the performance of the model trained on CIFAR-10. Our result in the main paper use Xavier Normal as our default initialzation. Previous works on PNs (Chrysos et al., 2022; 2023) have demonstrated the crucial role of appropriate parameter initialization in the final performance. In this section, we explore different parameter initialization methods for the linear layers in the model and train them on the CIFAR-10 dataset from scratch. To minimize the impact of confounding factors, all experiments are conducted without any data augmentation techniques or regularization methods. The results are shown in Table 18.

## N  ABLATION STUDY ON THE DESIGN OF MU-LAYER

Our original design in Eq. (1) factorizes $\mathbf{\Lambda}$, but not $\mathbf{A}$. One interesting question would be to explore factorizing also $\mathbf{A}$ as a low-rank decomposition. We follow the same idea as in $\mathbf{\Lambda}$ to factorize $\mathbf{A}$ as $\mathbf{A} = \mathbf{R} \cdot \mathbf{Q}$. The new design is visualized in Fig. 16.

We conduct an experiment on CIFAR10 to assess the performance of the model when both branches assume a low-rank decomposition. To provide a fair comparison with the main model, we leave the secondary branch as mentioned in the main paper. The results in Table 19 indicate that the original design performs favorably when compared to having low-rank decompositions in both branches.

Table 18: Comparison of Initialization Methods. The best performance is marked in **bold**. Note that * in the table uses PyTorch default initialization, it depends on the layer type. For linear layer of shape $(out, in)$, the values are initialized from $\mathcal{U}(-\sqrt{k}, \sqrt{k})$, where $k = \frac{1}{in}$.

| Initialization Method | Top-1 Acc (%) | Top-5 Acc (%) |
|---|---|---|
| Xavier Uniform (Glorot & Bengio, 2010) | 88.85 | 99.38 |
| Xavier Normal (Glorot & Bengio, 2010) | **89.13** | **99.85** |
| Kaiming Uniform (He et al., 2015) | 88.56 | 98.70 |
| Kaiming Normal (He et al., 2015) | 88.72 | 99.46 |
| Lecun Normal (LeCun et al., 2002) | 88.95 | 99.51 |
| Normal | 88.06 | 99.34 |
| Sparse (Martens et al., 2010) | 88.18 | 99.48 |
| Pytorch default* | 88.37 | 99.49 |

Table 19: Ablation study on the Mu-Layer design in Appendix N. Although considering the second branch as a low-rank decomposition performs generally worse, in some cases the difference is not large.

| | Baseline (ours) | Design 1 | Design 2 | Design 3 | Design 4 | Design 5 | Design 6 |
|---|---|---|---|---|---|---|---|
| The rank of Matrix RQ | - | 64 | 64 | 64 | 64 | 32 | 32 |
| The rank of Matrix BD | 64 | 32 | 64 | 16 | 8 | 16 | 8 |
| Acc | 88.1 | 87.1 | 85.2 | 86.4 | 86.2 | 86.1 | 85.6 |

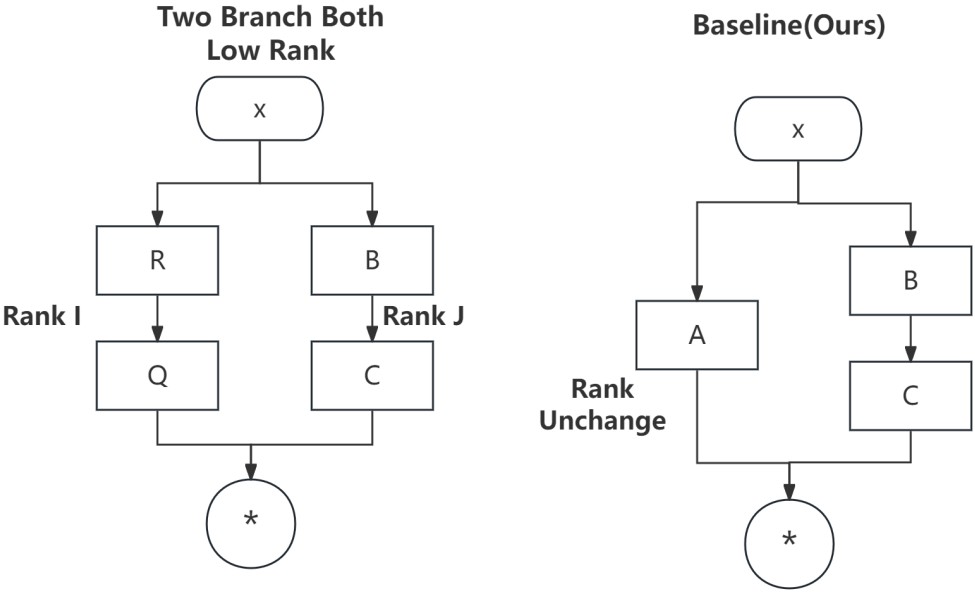

Figure 16: Different designs of Mu-Layer as studied in Appendix N.

