# OpenReview forum: "Multilinear Operator Networks"
_ICLR.cc/2024/Conference — ICLR 2024 poster_

### Official Review · Reviewer_6KA9 · 2023-11-02

**Soundness:** 3 good
**Presentation:** 3 good
**Contribution:** 3 good
**Rating:** 8
**Confidence:** 3

**Summary:**

This paper introduces a novel polynomial network called MONet, which exclusively relies on multilinear operators. The core layer of MONet is capable of capturing high-degree multiplicative channel-mixing operations for tokens, enabling non-linearity and, as a result, eliminating the need for activation layers. Extensive experiments were conducted on various image recognition and scientific computing benchmarks, demonstrating superior performance compared to previous polynomial networks and achieving results on par with state-of-the-art CNNs and Transformers.

**Strengths:**

1. The efficiency of this network design is truly impressive. In ImageNet, the Multi-stage MONet-S achieves performance comparable to the state-of-the-art, all without the use of any activations. It's also worth noting its robustness to image corruptions. When applied to solving ODEs, its performance appears strong based on the provided cases. The extensive experiments seem to highlight numerous promising characteristics of this new design, which warrant further exploration.
2. MONet simplifies prior polynomial networks by utilizing mu-layers. It appears reasonable that mu-layers are designed to accommodate only two-degree polynomials, as stacking these mu-layers enables high-order interactions.

**Weaknesses:**

1. In mu-layer, is it possible to use low-rank matrices multiplication for A? How will it perform
2. Is it feasible to employ low-rank matrix multiplication for A in the mu-layer? How would this alternative approach perform?
3. In the ODE experiments, could the authors offer more detailed quantitative results and compare them with other state-of-the-art methods, such as NeuralODE?
4. In the ODE experiment setup, is it necessary to use patch embeddings? Could you please provide additional details about the data processing and experimental settings for this specific task?

**Questions:**

1. In solving ODE, could the authors provide more details regarding how the model recovers the right-hand side of the Lotka-Volterra formula?

---

> ### Author Response · Authors · 2023-11-15
> **Response to Reviewer 6KA9**
>
> We are thankful to the reviewer 6KA9 for their effort to review our paper. We address below their concerns and questions:
>
> > Q1：Is it feasible to employ low-rank matrix multiplication for A in the mu-layer? How would this alternative approach perform?
>
> A1：We confirm that such a factorization is possible and we are thankful to the reviewer for suggesting this idea. We factorize the matrix $A$ of our model as $A = RQ$ of rank $k$. We try out different ranks $k$ and report the results below as conducted on a 6M parameter model trained on CIFAR10:
>
> |                    | Baseline(ours)   | Design 1 | Design 2 | Design 3 | Design 4 | Design 5 | Design 6 |
> | ------------------ | ---------------- | -------- | -------- | -------- | -------- | -------- | -------- |
> | Rank of matrix $RQ$   | - | 64       | 64       | 64       | 64       | 32       | 32       |
> | Rank of $BD$ | 64               | 32       | 64       | 16       | 8        | 16       | 8        |
> | Acc                | 88.1             | 87.1     | 85.2     | 86.4     | 86.2     | 86.1     | 85.6     |
>
> _______
>
> > Q2: In the ODE experiment setup, is it necessary to use patch embeddings? Could you please provide additional details about the data processing and experimental settings for this specific task?
>
> A2: We do not need patch embedding, since the data in the ODE model process time series. The setup for this experiment differs slightly from the image recognition setup, since now our input is a vector. Our data preprocessing uses standard practices in the ODE literature. For the experiment in sec. 4.4, we generated discrete data points using the underlying ground-truth formula. We add noise to the points and use those to train the model. Lastly, our model uses few Mu-layers, while the spatial shift is also not used in this case.
>
> _______
>
> > Q3: In solving ODE, could the authors provide more details regarding how the model recovers the right-hand side of the Lotka-Volterra formula?
>
> A3：We utilize MoNet to approximate the derivatives in our system of ordinary differential equations (ODEs). MoNet is specifically designed to output the rate of change of the state variables, namely dx/dt and dy/dt (i.e., the two rates of change that Lotka-Volterra includes). The architecture of MoNet generates a polynomial function of the input variables x and y, with the coefficients of this polynomial being determined by the learnable matrices W_i. Through the training process, MoNet learns the "optimal" values of these matrices W_i from the given data, effectively learning the dynamics described by the ODE. Once training has converged, the learned weights W_i are directly used to construct the specific forms of dx/dt and dy/dt.
>
> _______
>
> > Q4: More quantitative results and comparison with other ODE methods
>
> A4: We are thankful to the reviewer for the suggestion. Following existing ODE works quantitative evaluation, we conduct additional quantitative evaluation on image classification tasks (MNIST, CIFAR10 as in [1,2]). We follow the setting of [2] and conduct an image classification experiment. The model used consists of less than 0.2M parameters to make a fair comparison. The results in the table below indicate that  MoNet surpasses ANODE in CIFAR10 and SVHN, while it scores between NODE and ANODE in MNIST with accuracies larger than 96%.
>
>
> | Dataset | NODE      | ANODE     | PolyODE   |
> | ------- | --------- | --------- | --------- |
> | MNIST   | 96.4%±0.5 | 98.2%±0.1 | 97.2%±0.1 |
> | CIFAR10 | 53.7%±0.2 | 60.6±0.4  | 67.1%±0.2 |
> | SVHN    | 81.0%±0.6 | 83.5%±0.5 | 84.1%±0.5 |
>
> We show the same main evaluation tasks as [2] in appendix F to compare with those two works. We have presented training time and loss curves similar to those in the referenced work, and in the tasks provided by ANODE, our advantage lies in faster convergence and can restore to symbolic representation.
>
> If the reviewer has any remaining concerns, we would be happy to address them.
>
> ## References
>
> [1] Chen, et al. “Neural ordinary differential equations”, NeurIPS’18.
>
> [2] Dupont, et al. “Augmented neural odes, NeurIPS’19.

---

> > ### Author Response · Authors · 2023-11-21
> > **Are the concerns of the Reviewer 6KA9 addressed?**
> >
> > Dear reviewer 6KA9,
> >
> > We appreciate your feedback so far. Given that the discussion window is closing tomorrow, please let us know if you have any remaining concerns.
> >
> > Best,
> >
> > Authors

---

> > > ### Comment · Reviewer_6KA9 · 2023-11-23
> > >
> > > Thank you for your response. My concerns have been adequately addressed. I find it intriguing that the proposed MONet, even without activation functions, can surpass the performance of other polynomial networks. Consequently, I've decided to increase the score.

---

> > > > ### Author Response · Authors · 2023-11-23
> > > > **Thank you for your support!**
> > > >
> > > > Dear Reviewer 6KA9,
> > > >
> > > > We are truly delighted by your recognition of our contributions and your interest in our paper. Your feedback is invaluable for enhancing the quality of our manuscript. Many thanks！
> > > >
> > > > Best regards,
> > > >
> > > > The Authors

---

### Official Review · Reviewer_L73R · 2023-11-03

**Soundness:** 3 good
**Presentation:** 2 fair
**Contribution:** 3 good
**Rating:** 6
**Confidence:** 2

**Summary:**

The authors propose MONet, a Polynomial Network architecture that outperforms prior polynomial networks on image recognition tasks. The model uses stacked "multi-linear" operations to learn representations. The authors compare MONet models to state-of-the-art image classification models such as MLPMixer and ViT-16/B on image classification tasks, such as ImageNet1K and CIFAR10, along with medical image classification with MedMNIST and scientific computing with Neural ODE solver.

**Strengths:**

* Achieves impressive results on ImageNet1K compared to existing polynomial network work compared to vision transformer and CNN baselines, notably better particularly in Top-5 accuracy, which should be the more informative metric.
* Some good results in some of the categories for ImageNet-C under some image corruptions, such as weather and digital.
* Results are compared with state-of-the-art image classification baseline models such as MLPMixer and ViTs amongst others.
* Results are demonstrated on more than CIFAR-10/ImageNet with a tiny medical image classification dataset, and more interestingly, in the task of solving an ODE.
* FLOPS analysis is given (and it is reasonable) in the appendix, and is presented along with model size in parameters when comparing with models in e.g. Table 2. FLOPS/parameters appear reasonable in Table 1 compared to other models, however it seems perhaps VRAM usage is much more (see below), which is practically the bottleneck in most GPU-based training.
* One area where MONets may have more interpretability might be useful is in very specific cases, such as particular ODEs, that have a similar polynomial representation (as demonstrated in 4.4), but this seems like a very niche case.
* An Ablation analysis on the proposed methodology in the architecture is performed.

**Weaknesses:**

* This is a poorly written and organized paper as it is, which unfortunately lets down what appear to be some very good results, and perhaps an interesting method/model architecture.
* Virtually no motivation for the work aside from making polynomial networks have better generalization. We are given a sentence in the introduction "Interpretability and encryption..." with no reasons at all given as to why the proposed model architecture would be any more amenable to interpretation or encryption, and certainly no such results/analysis. Even the most academic work needs motivation. Getting closer to reaching state-of-the-art performance on some task alone is not enough of a motivation for a method if there are no tangible benefits over existing methods (and especially when it seems there are disadvantages in memory usage as covered below).
* Similarly, why are we interested in getting rid of activation functions? No motivation is given for this explicitly, and the only reason I can think of given the interpretability word is to remove non-linearity in the mapping from input to output. A sentence in the conclusion backs this up: "which avoids the requirement for activation functions or other non-linear mappings". However, the "multilinear" operation in this work is itself still a non-linear mapping of input to output (a polynomial of degree 2), making this statement false.
* The background is *way* too short and doesn't give near enough information on the current work's context in polynomial network literature.
* While the method is detailed enough, with much more detail in the appendix on some of the most interesting bits, the paper doesn't explain much at all what specifically is different from current polynomial networks, seemingly assuming the reader should know this. It is up to the authors to explain how their method fits into the current research literature in the background and method and compare their method explicitly with the closest existing work in literature.
* Requires 4 A100 GPUs to train for a batch size of 448 according to the authors (and they say they maximize batch size). This means that the maximum batch size on each GPU is 112. This is **much more VRAM usage** than the baselines being compared to that are achieving similar accuracy on ImageNet1K - for example, ViT B/16 can fit with a batch size of 256 on a single A100, and ResNet-50 a batch size of 512 on a single A100. Note it is not clear if these are 40GB or 80GB A100 models.
* Bold figures in captions are misleading, and do not it turns out always represent the best result as you might expect! For example, in Table 2

**Questions:**

* How is Mu-Layer novel compared to existing polynomial network architectures? Is the polynomial representation learned different, and why?
* What is the VRAM usage of the MONet models compared to baselines in Table 1, does it need much more VRAM in practice to train than those baselines?
* Why should be be interested in training ImageNet1K with MoNet instead of ViT or ResNet 50 which get comparable performance? In other words motivate your research work and method.
* Explain the statement "...which avoids the requirement for activation functions or other non-linear mappings" in the conclusion. How is a polynomial mapping linear, or am I misunderstanding this statement?
* Why are activation problems interesting to remove?
* How are MONets more interpretable in the general case?
* How are MONets more amenable to encryption than other models?

---

> ### Author Response · Authors · 2023-11-11
> **Response to Reviewer L73R (1/3)**
>
> We are thankful to the reviewer for their effort to review our paper. We address below their concerns and questions:
>
> > Q1: How is Mu-Layer novel compared to existing polynomial network architectures? Is the polynomial representation learned different, and why?
>
> A1: The difference from previous polynomial networks lies in our goal to design polynomial networks (PNs) that are competitive to the state-of-the-art networks *without relying on activation functions*. To the best of our knowledge, the core works that have explored PNs without activation functions are [2, 16, 17].
>
> To achieve that, we are inspired by the modern setup of considering the input as a sequence of tokens. The token-based input is widely used across a range of domains and modalities that last few years. The proposed modules (Mu-Layer and Poly-block) are customized to correspond to this form of data, which differs from the modules proposed previously. We note that this approach has not been considered by previous PNs. Furthermore, we propose additional modules, such as the pyramid patch embedding, which have not been considered before, especially in the context of polynomial networks.
>
> We highlight that a byproduct of our design (when compared to previous PNs) is that our method relies on matrix multiplications instead of convolutional layers. Arguably, this results in a weaker inductive bias as pointed out in the related works of MLP-based models. This can be particularly useful in domains outside of images, e.g., in the ODE experiments.
>
> ____
>
> > Q2: What is the VRAM usage of the MONet models compared to baselines in Table 1, does it need much more VRAM in practice to train than those baselines?
>
> A2: We are thankful to the reviewer for helping us clarify this potential misunderstanding. **MONet does not consume more VRAM**. We kindly remind the reviewer that the reported batch size in the paper without “the total batchsize” usually refers to the batch size used per single GPU, and this terminology is commonly employed in similar works as a common practice [1-3].
>
> However, to avoid any further misunderstanding, we have changed the text to refer explicitly to the batchsize “per GPU”. In our experiments, we use a batchsize of 448 per GPU, which is larger than the batchsize of 256 the reviewer reports for ViT. Overall, the total batchsize of 1792 is used in our case.
>
> ____
>
> > Q3: Why should we be interested in training ImageNet1K with MoNet instead of ViT or ResNet 50 which get comparable performance? In other words, motivate your research work and method.
>
> A3: We respectfully disagree with the reviewer. The community does value diverse architectures and in fact much of the progress observed can be (partly) attributed to the architecture design, which manifests a different inductive bias on the task at hand.
> In this work, we propose an orthogonal direction to neural architecture design that does not follow the pattern of stacking linear/convolutional layers and non-linear activation functions. We believe that by showing that such an alternative can perform better or on par with standard baselines, we contribute a valuable direction in the search for the optimal deep architecture. As such, even though, as Table 2 suggests, our model outperforms both ResNet 50 and ViT, we believe our contribution lies beyond this improvement in performance.
> ____
>
> > Q4: Non-linear mapping and multilinear operations.
>
> A4: The mapping we use is multilinear. We use the standard terminology as used in the tensor literature that extends beyond the field of machine learning. The links between multilinear algebra and signal processing have long been established and use a standard terminology, as can be verified in [12-14]. For instance, an interested reader can find further information on the seminal review paper of Kolda and Bader [11].
>
> However, in the revised text we clarify in the introduction that we are using this multilinear terminology and provide links for the interested reader.
>
> In addition, to avoid any misunderstanding, we have also clarified the statement in the conclusion that we can avoid the activation functions, instead of all non-linear mappings.

---

> > ### Comment · Reviewer_L73R · 2023-11-22
> >
> > I'd like to thank the authors for their rebuttal, and apologize for my late participation in the rebuttal period - this was due to exceptional circumstances beyond my control.
> >
> > Q1. This is good, and should be in the paper. Unfortunately I see that the revised paper doesn't seem to include it?
> >
> > Q2. This is good to clarify - it wasn't obvious at all before, but I see you have updated the paper.
> >
> > Q3. I respectfully disagree - and please don't misunderstand, I'm all for exploring alternative methods of doing something, even if the performance is not better. Even so, you have to motivate your method outside of it **just being different**, this as fundamental a part of writing a scientific paper as the method and results. While you demonstrate an extremely narrow case in which your method is more interpretable, showing additional benefits that were more general would substantially better motivate the work.
> >
> > Q4. Thanks for clarifying this, and for updating the paper to remove the linear claims.

---

> ### Author Response · Authors · 2023-11-11
> **Response to Reviewer L73R (2/3)**
>
> > Q5: Why are activation problems interesting to remove?
>
> A5: There are two aspects that removing activation functions has potential benefits: 1) in the theoretical analysis of deep networks and 2) in (potential) real-world applications. Please let us provide few usecases:
>
> 1.  The analysis of neural networks and a deeper understanding of their dynamics and generalization still remains elusive. This can be partly attributed to the use of activation functions, as identified explicitly in [6]. The workaround is that sometimes activation functions are removed, e.g., in the popular papers of [7-9], which results in a linear approximation in the case of neural networks. Instead, in polynomial networks (PNs), the activation functions are not a strict requirement, since the PNs can approximate distributions without them as guaranteed by the Stone-Weierstrass theorem. In addition, Zhu et al [10] show that obtaining the sample complexity is significantly simplified in the absence of activation functions.
> Lastly, recent works even consider removing some of the activation functions in traditional neural networks, e.g. [15], in order to understand the expressivity of standard models and increase our understanding of deep neural networks.
>
> 2. A number of real-world applications require the use of networks without activation functions, e.g., in encryption, privacy-preserving settings or in interpretability. In such cases, the activation functions do pose challenges in the use of encryption protocols such as Fully Homomorphic Encryption (LFHE).  LFHE can only use addition and/or multiplication operations and as such they cannot use traditional neural networks. However, we note that experimenting in this direction is beyond the scope of our work.
> Furthermore, the benefits of interpretability in the absence of activation functions have already been studied in [4, 5]. In [5], the authors study interpretability from the lens of recovering the equations governing a dynamical system. Dynamical systems and ODEs are important in scientific fields in biology, chemistry and related fields. This work highlights that this is not possible in the presence of activation functions. In our work, we approach interpretability in section 4.4 through the same lens.
> ____
>
> > Q6: How are MONets more interpretable in the general case?
>
> A6：In this work, we focus on interpretability from the lens of recovering the ground-truth function (e.g., in a dynamical system) when this can be expressed as a polynomial expansion, as defined in [5]. In that sense, we exhibit in sec. 4.4 how the proposed method can recover the equations behind the dynamical system. We understand that this may differ from other definitions the reviewer might have in mind and we are happy to discuss this further.
>
> ____
>
> > Q7: How are MONets more amenable to encryption than other models?
>
> A7: The Leveled Fully Homomorphic Encryption (LFHE), can provide a high level of security for sensitive information. The core limitation of FHE (and especially LFHE) is that they support only addition and multiplication as operations. That means that traditional neural networks cannot fit into such privacy constraints, making developments in MLP-Mixer and similar models invalid for many real-world applications. On the contrary, the proposed model relies on those two operations, i.e., addition and multiplication, thus making it amenable to encryption.
> ____
>
>
> > Q8: Bold figures in captions do not always represent the best result, e.g., in Table 2.
>
> A8: We respectfully disagree with the reviewer. As typically done in the literature, we conduct experiments in different categories of architectures. The categories are often determined by the number of parameters, where the “small” category uses less than 15 million parameters, medium models with less than 40 million, while the “large” category exceeds 40 million parameters.
>
> However, to increase the readability of our paper, we have indicated the small models in green color, medium models in red color and the larger models in blue color.
>
>
> ____
>
> > Q9: Background work is way too short.
>
> A9: We appreciate the remark from the reviewer and the effort to improve our work. We have included a new section in the appendix (sec. B in the revised manuscript), where we provide additional background specifically for polynomial networks.
>
> If the reviewer has any remaining concerns, we are happy to provide further clarifications. The revised text is denoted in red for readability.

---

> > ### Comment · Reviewer_L73R · 2023-11-22
> >
> > Q5. I do understand that removing non-linear activation functions leaves a linear model which is more interpretable, but as we discussed already the multilinear networks used here are not linear, so that is not relevant as far as I can see. I also understand that Stone-Weierstrass means polynets can be universal function approximations, however this also doesn't seem to answer why removing activation functions itself is a goal. I'm not familiar with the works [10, 15], but I'll assume they have some relevance to polynets?
> >
> > I think the best motivation you gave above could be on the applications that require models without activation functions, and I would perhaps focus on that as an example of why we should be interested in removing activation functions. It is relatively clear and concise, and a much stronger argument than your claims that interpretability is better.
> >
> > To be clear, the reason I asked this question in the initial review is that "removing activation functions" is used repeatedly in the paper as a goal that should be self-apparent. However, this is clearly far from the case, you need to motivate this better or focus on it less.
> >
> > Q6. My question was how they are more interpretable in general. The case presented is extremely narrow, where it happens the dynamical system has the very same form as the multilinear network being trained - this cannot be a general situation, and does little to convince the reader that the model is more interpretable in any general sense.
> >
> > Q7. This is good - and again I think you should consider motivate your work using this example!
> >
> > Q8. Thanks - I do believe the general consensus is that bold numbers are best in a table, so this will be less misleading, even if it was unintentional.
> >
> > Q9. While it is great that a section has been added with more background in the appendix, to make your paper more reasonable to a general audience I believe it is imperative you include this in the main body of the paper.

---

> > > ### Author Response · Authors · 2023-11-22
> > > **Response to the new comments of Reviewer L73R**
> > >
> > > Dear reviewer L73R,
> > >
> > > We are truly thankful for the detailed responses and the appreciation of our rebuttal. We incorporate all the suggestions in the revised manuscript; those changes are denoted in red for clarity. We clarify below your questions (following the same numbering):
> > >
> > >
> > > > Q1. Where is the response (on the comparison with prior PNs) in the paper?
> > >
> > > A1: We include the revised text at the last paragraph of sec. 2 (i.e. related work). [Here](https://imgur.com/a/lZ1vXSP) is a printscreen of this paragraph.
> > > _______
> > >
> > > > Q3. What is your motivation?
> > >
> > > A3: We are thankful for the suggestion on basing the motivation around the potential applications (as suggested in Q5 and Q7). We have updated the introduction accordingly.
> > > _______
> > >
> > > > Q5: Soften the lack of activation functions
> > >
> > > As mentioned above, we opt to motivate our work from the application side. However, to make the manuscript clearer, we removed the claim on “removing the activation functions” from a) the bullet points in the introduction, b) the related work and c) the experimental section, thus softening the lack of activation functions as requested.
> > > _______
> > >
> > > > Q6. The case (on ODEs) presented is extremely narrow.
> > >
> > > A6: We agree with the reviewer that our claim covers only the dynamical systems that follow a polynomial form, e.g., the  Lotka-Volterra model.  In this case, the proposed model does provide the additional benefit of interpretability.
> > >
> > > Nevertheless, to increase the clarity of our paper, we remove the related claim from the introduction, while we soften our related claim (in the ODE section in sec. 4.3) as follows:
> > >
> > > An additional advantage of our method is the ability to model functions that have a polynomial form (e.g, in ordinary differential equations) in an interpretable manner.
> > > _______
> > >
> > > > Q7. Perhaps you should consider motivating your work with this example.
> > >
> > > A7: We are thankful to the reviewer for the recommendation. We have directly modified the first two paragraphs of the introduction to use this as a motivating example.
> > > _______
> > >
> > > > Q9. The background work on polynomials should be in the main paper.
> > >
> > > A9: We appreciate the suggestion from the reviewer. We have rewritten the paragraphs of sec. 2 for the polynomial networks, such that it includes the background information requested.
> > >
> > > Once again, we are grateful to the reviewer for their suggestions and helping us improve our manuscript. If there are any remaining concerns, we would be happy to address them.

---

> ### Author Response · Authors · 2023-11-11
> **Response to Reviewer L73R (3/3)**
>
> ## References
>
> [1] Touvron, et al. “Resmlp: Feedforward networks for image classification with data-efficient training”, T-PAMI’22.
>
> [2] Chrysos, et al. “Deep polynomial neural networks”, T-PAMI’21.
>
> [3] Wang, et al. “Pyramid vision transformer: A versatile backbone for dense prediction without convolutions”, ICCV’21.
>
> [4] Dubey, et al. “Scalable interpretability via polynomials”, NeurIPS’22.
>
> [5] Fronk and Petzold. “Interpretable polynomial neural ordinary differential equations”, Chaos: An Interdisciplinary Journal of Nonlinear Science, 33(4), 2023.
>
> [6] Arora, et al. "A convergence analysis of gradient descent for deep linear neural networks", ICLR'19.
>
> [7] Saxe, et al. "Exact solutions to the nonlinear dynamics of learning in deep linear neural networks", ICLR'14.
>
> [8] Laurent and Brecht. “Deep linear networks with arbitrary loss: All local minima are global”, ICML’18.
>
> [9] Lampinen and Ganguli. “An analytic theory of generalization dynamics and transfer learning in deep linear networks”, ICLR’19.
>
> [10] Zhu, et al. “Controlling the complexity and lipschitz constant improves polynomial nets”, ICLR’22.
>
> [11] Kolda and Bader. “Tensor decompositions and applications”, SIAM review, 2009.
>
> [12] De Lathauwer and De Moor. “From matrix to tensor: Multilinear algebra and signal processing”, in Mathematics in Signal Processing IV, Oxford, 1998.
>
> [13] De Lathauwer. “A link between the canonical decomposition in multilinear algebra and simultaneous matrix diagonalization”, SIAM journal on Matrix Analysis and Applications 28.3 (2006).
>
> [14] Sidiropoulos and Bro. “On the uniqueness of multilinear decomposition of N‐way arrays”, Journal of Chemometrics, 2000.
>
> [15] Mehmeti-Göpel and Disselhoff. “Nonlinear Advantage: Trained Networks Might Not Be As Complex as You Think”, ICML’23.
>
> [16] Chrysos, et al. “Augmenting Deep Classifiers with Polynomial Neural Networks”, ECCV’22.
>
> [17] Chrysos, et al. “Regularization of polynomial networks for image recognition”, CVPR’23.

---

> ### Author Response · Authors · 2023-11-16
> **Are the concerns of the Reviewer L73R addressed?**
>
> Dear reviewer L73R,
>
> We appreciate your feedback and concerns on our work. We strive to address your questions in our previous responses. The key concerns were the VRAM usage, the novelty over previous methods and the interest in removing activation functions. Our response above addresses those concerns, clarifying that our model does *not* use more VRAM and expand on the novelty and the interest in removing activation functions. Regarding the clarity of the presentation, we strived to explain the method in a clear and understandable fashion (Reviewer kEbz identifies that "the theory underlying the layer is well-proven and explained."). Nevertheless, if there are specific parts that are unclear to the reviewer, we are more than happy to revise them.
>
> Please let us know if you have any remaining concerns.
>
> Best,
>
> Authors

---

> > ### Author Response · Authors · 2023-11-21
> > **Are there any remaining concerns of the Reviewer L73R?**
> >
> > Dear reviewer L73R,
> >
> > We appreciate your feedback so far. Given that the discussion window is closing tomorrow, please let us know if you have any remaining concerns. Otherwise, we would appreciate it if you increase your score to reflect our revised manuscript.
> >
> > Best,
> >
> > Authors

---

> ### Author Response · Authors · 2023-11-23
> **Are there any remaining concerns of the Reviewer L73R?**
>
> Dear reviewer L73R,
>
> We appreciate your time and effort to provide invaluable suggestions regarding the writing and motivation. Given the extensive revisions to the paper (according to your suggestions), we would be grateful if you reconsider your score to reflect our revised manuscript.
>
> Best,
>
> Authors

---

### Official Review · Reviewer_kEbz · 2023-11-03

**Soundness:** 2 fair
**Presentation:** 2 fair
**Contribution:** 2 fair
**Rating:** 6
**Confidence:** 3

**Summary:**

The authors propose a multilinear operator network which avoids nonlinear interactions. The architecture makes use of "Poly-blocks" composed of "Mu layers" which can capture up to fourth degree multiplicative interactions in input tokens. The authors demonstrate that their multilinear networks are competitive against other polynets, traditional MLPs, and ResNets on tasks such as image classification and ODE problems.

**Strengths:**

1. The Mu layer design is simple enough and the theory underlying the layer is well-proven and explained.
2. The MONets appear to be quite competitive against a variety of baselines on a wide array of tasks.
3. The number of FLOPs and memory required to operate the MONet are drastically reduced compared to other baselines.

**Weaknesses:**

1. Fundamentally, it's unclear why the MONet should generalize well to other tasks if the Poly-block is only capturing up to fourth degree interactions. For instance, for general MLPs, results such as Cybenko's theorem tells us the MLP should capture many families of Lebesgue integrable functions but here the limit is restricted to functions that are captured by compositions of fourth degree approximations and skip connections. The surprising capacity of the MONet is not adequately discussed.

2. The numbers reported for the P-Nets [1] do not appear to be accurate. With >11 parameter P-Nets, the achievable accuracy on CIFAR-10 is 94.5% and ImageNet ~77% Top-1 accuracy/ ~94% Top-5 accuracy. The exact accuracies reported in [1] would either closely-defeat or match the performance of the MONets with similar or drastically fewer numbers of parameters. Please address this discrepancy.


[1] Chrysos, G. G., Moschoglou, S., Bouritsas, G., Deng, J., Panagakis, Y., & Zafeiriou, S. (2021). Deep polynomial neural networks. IEEE transactions on pattern analysis and machine intelligence, 44(8), 4021-4034.

**Questions:**

Please see weaknesses. Is there some intentional restriction placed on the P-Net method? I believe you have pulled the numbers from [2], which is also in discrepancy with the first work (by the same authors, no less). Similarly, the ResNet 50 numbers are reported strangely. It can be easily fine-tuned to achieve 78.8%/94.5% Top1/5 accuracies, respectively, but this is not in concordance with Table 2.

[2] Chrysos, G. G., Wang, B., Deng, J., & Cevher, V. (2023). Regularization of polynomial networks for image recognition. In Proceedings of the IEEE/CVF Conference on Computer Vision and Pattern Recognition (pp. 16123-16132).

---

> ### Author Response · Authors · 2023-11-11
> **Response to Reviewer kEbz**
>
> We are thankful to the reviewer kEbz for their effort to review the paper. We clarify the questions raised by the reviewer:
>
> > Q1: Poly-block is only capturing up to fourth degree interactions which limit its capacity. The surprising capacity of the MONet is not adequately discussed.
>
> A1: We thank the reviewer for the remark. Let us clarify why the total network results in a higher-degree expansion. We do mention in the text that the Poly-block captures up to 4th degree interactions as the reviewer mentions. However, we also note that a single network is composed of several Poly-blocks, usually tens of those.
>
> *Simple case with two Poly-blocks*: The output of the first Poly-block (which is 4th degree interactions of its inputs) of the network is directly an input to the next Poly-block. The output of the second Poly-block is again 4th degree interactions of the output of the first block (which was already 4th degree of the inputs). Therefore, the output of the second Poly-block with respect to the input (e.g., image) captures up to 16th degree interactions.
>
> To generalize from two Poly-blocks to a sequence of those as used in our experiments, we notice that the output of $N$ sequential blocks is $4^N$. Given than $N>10$, we assume that this is a sufficient number to capture high-degree interactions. In practice, we observe that in our experiments this degree is sufficient for a competitive accuracy.
>
> Nevertheless, we agree with the reviewer that it remains an open and interesting problem of identifying the degree of interactions required by the total network to tackle each task. We have inserted a related note in sec. 3 to inform the reader, since we do believe this is an interesting future step. The note is marked with red color for visual clarity.
>
> > Q2: Is there some intentional restriction placed on the $\Pi-$Nets?
>
> Please let us clarify. **There are no restrictions placed on $\Pi-$Nets**.
> However, there is a difference in the experimental settings. The following models are used:
> $\Pi-$Net (Table 2): This refers to the small model (equivalent to ResNet-18) WITHOUT activation functions.
> Hybrid $\Pi-$Net (Table 2): This is also a small model WITH activation functions.
>
> We elaborate on these points below:
> The numbers referred to by the reviewer (e.g., 94.5% on cifar10) are reported by the authors when **activation functions are used**. For instance, in the ImageNet experiment in [1] (page 7, sec. 4.2) they identify that “the second order of each residual block is normalized with a hyperbolic tangent unit”, i.e., a tanh activation function is used in each block. However, if we compare with $\Pi-$Nets without activation functions, then we obtain those scores reported in our Table. In fact, the authors of [1] have conducted some of those experiments in the supplementary material, e.g., in Table 1 [here](https://openaccess.thecvf.com/content_CVPR_2020/supplemental/Chrysos_P-nets_Deep_Polynomial_CVPR_2020_supplemental.pdf). This is also explicitly identified by the authors of [1] in the code release of the paper. In this [table](https://github.com/grigorisg9gr/polynomial_nets/tree/master/classification-NO-activation-function#imagenet-1k) they elaborate on the accuracy reached when using activation functions and when you do not use activation functions. In conclusion, *there is no restrictions placed  on our reported results on $\Pi-$Nets; we simply report $\Pi-$Nets without activation functions*.
>
> Note that our original manuscript identified that $\Pi-$Nets do not use activation functions in Table 2 (in the second-to-last column). The models with activation functions were notified as `hybrid $\Pi-$Nets’. However, to clarify this point even further, we have inserted a footnote that explicitly identifies that we are using the $\Pi-$Nets without activation functions.
>
> Furthermore, the accuracy of 77% on ImageNet (in [1]) involves a ResNet-50 model and the equivalent $\Pi-$Net WITH activation functions. That would be under the category of medium models in our case, where the accuracy of our model is 81.3%.
>
> > Q3: ResNet 50 numbers in Table 2 are reported strangely.
>
> A3: We use the accuracy as reported in [A], for a fair comparison. To avoid any confusion, we have replaced this with the accuracy reported in the original ResNet paper (now the accuracy on Table 2 is 77.2%). We do agree that additional techniques, such as finetuning, can improve the results further, however we would need to apply similar techniques to all methods for a fair comparison. Our evaluation is the standard and fair evaluation for all different architectures.
>
> Nevertheless, we are thankful for the suggestion. To increase the transparency on the reported numbers, we added sec. K.3 (appendix), where we explain how we obtained the core compared results on the ImageNet experiment.
>
> If the reviewer has any remaining questions or concerns, we are happy to address them.
>
> [A] S2-mlp: Spatial-shift mlp architecture for vision, WACV'22.

---

> ### Author Response · Authors · 2023-11-16
> **Are the concerns of the Reviewer kEbz addressed?**
>
> Dear reviewer kEbz,
>
> We appreciate your feedback and concerns. We strive to address your questions in our previous responses. The key concerns were the degree of expansions and the results reported for the baseline of $\Pi-$Nets.
>
> Please let us know if you have any remaining concerns.
>
> Best,
> Authors

---

> > ### Author Response · Authors · 2023-11-21
> > **Are there any remaining concerns of the Reviewer kEbz?**
> >
> > Dear reviewer kEbz,
> >
> > We appreciate your feedback so far. Given that the discussion window is closing tomorrow, please let us know if you have any remaining concerns. Otherwise, we would appreciate it if you increase your score to reflect our revised manuscript.
> >
> > Best,
> >
> > Authors

---

> ### Comment · Reviewer_kEbz · 2023-11-23
> **Response to Authors**
>
> I want to thank the authors for their responses to my remarks. Especially regarding the supposed disparity of results reported in [1]; the reported numbers are now clear to me.
>
> I do understand that the composition of poly-blocks results in capturing interactions up to $4^N$ degree, where $N$ is the number of poly-blocks. However, citing "interpretability" as a driving reason for the usage of MONets (and PolyNets in general) is not a strong enough motivation for most tasks since although we're guaranteed a polynomial approximation of any nonlinear function by the very general Weierstrass approximation theorem, functionally this degree could be incredibly high. Consequently, there's very little interpretability, other than in the case of ODE approximation, where a polynomial expansion can reveal a great about dynamics, and thus the motivation is rather weak.
>
> However, I do believe that the posted revisions do illuminate several of the questions and concerns of the reviewers, including my own, and I am increasing my score to a marginal accept.

---

> ### Author Response · Authors · 2023-11-23
> **Thank you for your support!**
>
> Dear Reviewer kEbz,
>
> We are delighted by your recognition of our contributions. Your feedback is invaluable for improving our manuscript. Once again, we are thankful for your feedback. Upon the feedback during the discussion period, we have softened our claim on interpretability and we appreciate your feedback on this topic. We agree that this remains an interesting topic to explore and we will write this explicitly in the revised version of the paper. If you have any additional feedback, it is welcome.
>
>
> Best regards,
>
> The Authors

---

### Meta-Review · Area_Chair_LsQZ · 2023-12-12

**Metareview:**

This paper introduces a type of polynomial network based on multilinear operators. The core idea of MONet is to build a network based on multiplicative interactions of the inputs with one another.

Strengths of this work are a novel framing of polynomial networks and strong empirical results on various benchmark datasets, including comparison with strong baselines. The authors and reviewers also did a great job discussing and updating the manuscript, which made it stronger and finally all reviewers were convinced by the merit of this work and the quality of the presentation.

A weakness of this paper are remaining concerns around the benefit of removing activation functions, it is not fully clear why this is a goal in and of itself, but generally there is agreement with the authors that novel architectures do drive surprising progress in many cases.

Overall, the presentation of the paper is now in good shape and the quality of the work and empirical analysis itself warrants publication at ICLR.

**Justification For Why Not Higher Score:**

While this paper improves performance of polynomial networks and benchmarks MONets widely, there is no breakthrough or strong reason to move this paper to an oral or spotlight presentation.

**Justification For Why Not Lower Score:**

Reviewers all agree, that the paper merits acceptance to the conference.

---

### Decision · Program_Chairs · 2024-01-16

Accept (poster)